# The multi-subunit GID/CTLH E3 ubiquitin ligase promotes cell proliferation and targets the transcription factor Hbp1 for degradation

Fabienne Lampert[1][†]*, Diana Stafa[1][†][‡], Algera Goga[2], Martin Varis Soste[1][§], Samuel Gilberto[1], Natacha Olieric[3], Paola Picotti[1], Markus Stoffel[2], Matthias Peter[1]*

[1]Institute of Biochemistry, ETH Zürich, Zürich, Switzerland; [2]Institute of Molecular Health Sciences, ETH Zürich, Zürich, Switzerland; [3]Laboratory of Biomolecular Research, Division of Biology and Chemistry, Paul Scherrer Institute, Villigen, Switzerland

*For correspondence:
fabienne.lampert@bc.biol.ethz.ch
(FL);
matthias.peter@bc.biol.ethz.ch
(MP)

[†]These authors contributed equally to this work

Present address: [‡]Department of Biochemistry and Molecular Biology, Tulane University School of Medicine, New Orleans, United States; [§]Donnelly Centre for Cellular and Biomolecular Research, University of Toronto, Toronto, Canada

Competing interests: The authors declare that no competing interests exist.

**Abstract** In yeast, the glucose-induced degradation-deficient (GID) E3 ligase selectively degrades superfluous gluconeogenic enzymes. Here, we identified all subunits of the mammalian GID/CTLH complex and provide a comprehensive map of its hierarchical organization and step-wise assembly. Biochemical reconstitution demonstrates that the mammalian complex possesses inherent E3 ubiquitin ligase activity, using Ube2H as its cognate E2. Deletions of multiple GID subunits compromise cell proliferation, and this defect is accompanied by deregulation of critical cell cycle markers such as the retinoblastoma (Rb) tumor suppressor, phospho-Histone H3 and Cyclin A. We identify the negative regulator of pro-proliferative genes Hbp1 as a *bonafide* GID/CTLH proteolytic substrate. Indeed, Hbp1 accumulates in cells lacking GID/CTLH activity, and Hbp1 physically interacts and is ubiquitinated in vitro by reconstituted GID/CTLH complexes. Our biochemical and cellular analysis thus demonstrates that the GID/CTLH complex prevents cell cycle exit in G1, at least in part by degrading Hbp1.
DOI: https://doi.org/10.7554/eLife.35528.001

## Introduction

Cellular proteostasis involves the coordinated and compensatory action of pathways that control biogenesis, folding, trafficking and breakdown of proteins allowing the cell to adapt to physiological or pathological environmental changes. A hallmark of proteome balance is the ubiquitin-proteasome system (UPS) that degrades roughly 80% of multi-ubiquitinated proteins (*Collins and Goldberg, 2017*). In humans, about 600 distinct E3 ubiquitin ligases mark short-lived, malfunctioning and toxic molecules for degradation by 26S proteasomes (*Metzger et al., 2012*). Consistent with the fundamental role of this major cellular quality control nexus, perturbations within the network are associated with many human diseases, including cancer, neurodegeneration and viral infections (*Huang and Dixit, 2016*; *Metzger et al., 2012*). The UPS is also required for cells to adjust to different nutrient conditions such as limiting carbon sources. Indeed, changing metabolic flux is often controlled by regulating the relative abundance of rate-limiting enzymes that function in distinct exergonic pathways (*Nakatsukasa et al., 2015*). Particularly in yeast, an organism where gluconeogenesis and glycolytic activity are intermittently coordinated, the multi-subunit GID E3 ligase complex specifically targets the surplus of gluconeogenic enzymes, including the conserved Fructose-1,6-bisphosphatase 1 (Fbp1), for proteasomal degradation. Adequate glucose levels trigger the

enzymatic activity of the complex by either inducing the stabilization and/or de novo synthesis of its critical subunit Gid4 (*Santt et al., 2008*). These original findings were broadened by a recent study, suggesting the presence of a new GID-based constitutive proteolytic pathway termed the Pro/N-end rule pathway, wherein N-terminal proline residues and a subsequent stretch of partially flexible amino acids provide the signature for GID-E3 substrate recognition (*Chen et al., 2017*). Although the generality of such a model is tempting, to date known GID substrates are limited to enzymes that switch on glucose production in the presence of non-fermentable carbon sources. In mammals, gluconeogenesis is almost exclusively performed by hepatocytes but also takes place, albeit to a much lower extent, in the kidney cortex and intestine (*Mutel et al., 2011*; *Previs et al., 2009*). Indeed, maintenance of blood glucose levels during fasting/starvation and/or rigorous exercise is primarily achieved through the breakdown of liver glycogen stocks (glycogenolysis) and the conversion of non-sugar carbon substrates via gluconeogenesis (*Rui, 2014*). Thus, while the biochemistry of many metabolic pathways is largely equivalent from bacteria to humans, the biological relevance and control mechanisms have adapted to the needs of the respective organisms (*Otterstedt et al., 2004*). Nevertheless, all seven yeast GID proteins have homologous proteins in humans. Some of them, RanBP9 (Gid1), Rmnd5 (Gid2), Armc8 (Gid5), Twa1 (Gid8) and MAEA (GID9), have been found together in an ubiquitously expressed high-molecular weight assembly localized both to the nucleus and cytoplasm, also referred to as the C-terminal to LisH (CTLH) complex, after a sequence motif shared between several subunits (*Kobayashi et al., 2007*). However, neither the counterpart of the critical yeast regulatory subunit Gid4, c17orf39, nor the Gid7 sequence homologue, WDR26, were identified within the mammalian complex (*Francis et al., 2013*; *Kobayashi et al., 2007*). Instead, Mkln1, a protein involved in cell adhesion and the regulation of cytoskeleton dynamics, was suggested to replace WDR26 in humans (*Delto et al., 2015*; *Francis et al., 2013*). In turn, the WD40-containing protein WDR26 was previously identified as a substrate adaptor for the large family of Cullin4-RING ubiquitin ligases (CRL4s) (*Higa et al., 2006*; *Piwko et al., 2010*). Overall, the majority of the mammalian GID/CTLH complex components are poorly characterized, with the exception of the non-canonical member of the Ran-binding protein family RanBP9 (*Salemi et al., 2017*). RanBP9 was reported to integrate into multiple protein complexes, has numerous interaction partners and was proposed to regulate a diverse set of cellular processes ranging from transcriptional regulation, cell morphology and cell cycle progression to apoptosis (*Atabakhsh et al., 2012*; *Salemi et al., 2017*; *Suresh et al., 2012*). However, to date a comprehensive analysis of the mammalian GID complex as a functional entity and putative E3 ligase is still missing (*Salemi et al., 2017*). Furthermore, none of the human GID proteins have been studied in the context of a potentially conserved role in negatively regulating gluconeogenesis in mammals. We have a longstanding interest in the function and regulation of CRL4 ligases (*Brodersen et al., 2016*; *Mouysset et al., 2015*), and WDR26 was identified as a presumptive CRL4 adapter protein, possibly involved in cell cycle regulation (*Piwko et al., 2010*). However, it remains to be examined whether WDR26 indeed functions as a CRL4 adaptor in this context or alternatively acts as the sequence homologue of Gid7 incorporated in the putative GID/CTLH E3 ligase complex. Interestingly, using a combinatorial approach of systematic AP-MS, biochemical reconstitution and cellular phenotyping, we discovered that WDR26 is a stable subunit of the human GID/CTLH complex. Moreover, we identified additional human GID subunits and demonstrate that the E3 ubiquitin ligase activity of this multi-subunit complex is evolutionary conserved. Finally, we provide evidence that cellular GID activity is critical for maintenance of normal cell proliferation rates, possibly by direct ubiquitination of the transcription factor Hbp1.

## Results

### Comparative proteomics identifies ten core components of the human GID/CTLH complex

To investigate whether the putative CRL4 adapter WDR26 contributes to cellular fitness, we first deleted WDR26 from RPE cells using lentiviral transduction of the CRISPR-Cas9 system and observed a significant growth inhibition phenotype in these cells as judged by clonogenic and MTT assays (*Figure 1A and B*). To clarify whether the proliferation defect following WDR26 perturbation is attributable to the loss of a specific Cul4-based E3 ligase function, we used proteomics to examine the global WDR26-interactome. Interestingly, the data exposed WDR26 as a core part of a protein

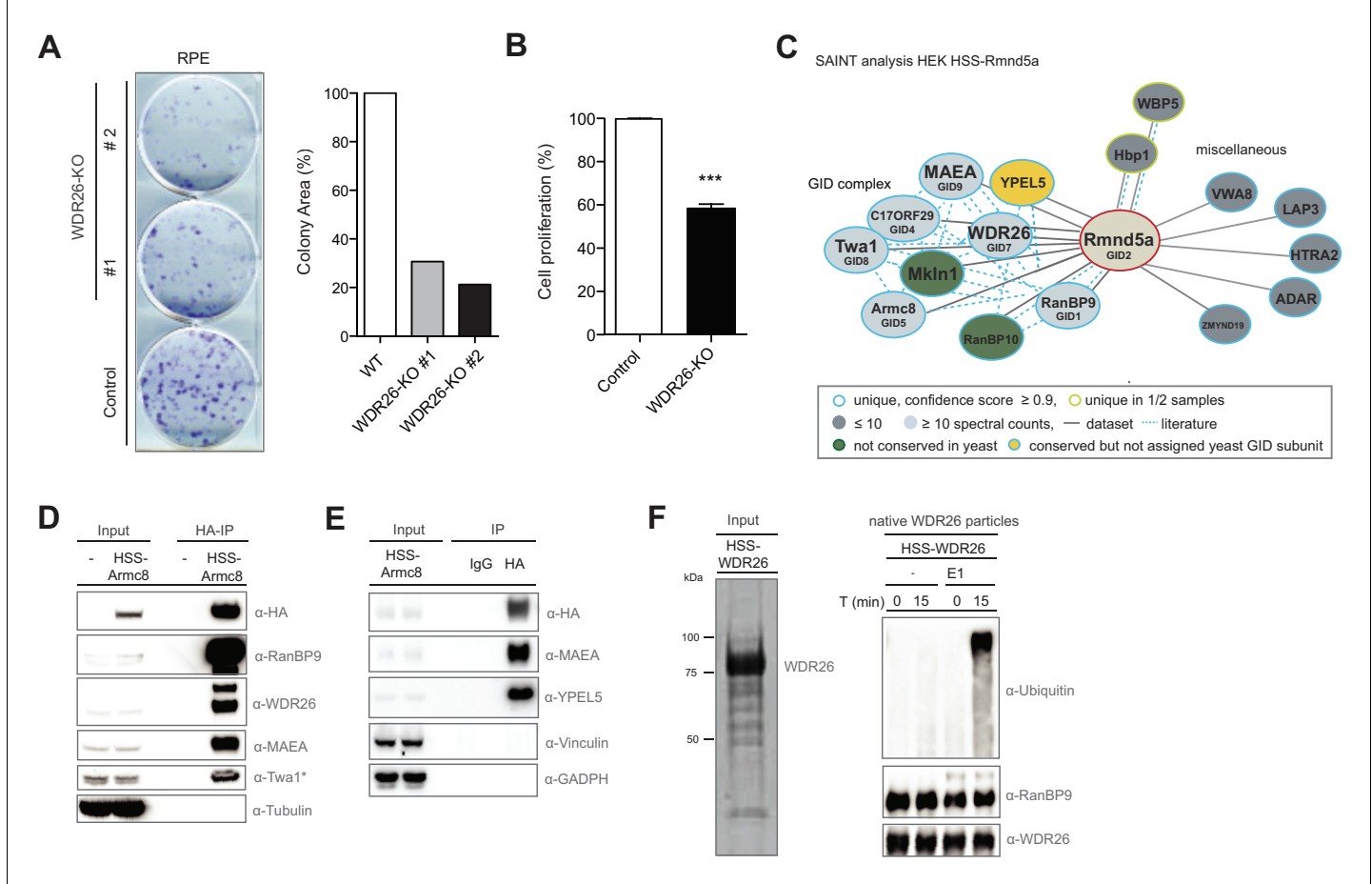

**Figure 1.** The human GID complex is composed of 10 subunits and possesses E3 ligase activity. (A) Representative clonogenic survival assay and corresponding quantification of RPE cells (technical replicates, n ≥ 2) treated with lentiviruses expressing either control gRNAs (WT) or two gRNAs deleting the GID subunit WDR26 (WDR26-KO). (B) Cell proliferation of RPE control and WDR26-KO cells was quantified by MTT assays between days 6–9 after lentiviral transfection. Data are shown as mean of quadruplicates and % change in signal relative to control gRNA-treated cells ± SD, n = 3, ***p≤0.0003. (C) Network of Rmnd5a high-confidence interacting proteins (HCIPs) in HEK-239 cells analyzed by SAINT (confidence score ≥0.9, FC ≥2, n = 2). GID subunits with no sequence/functional homologue within the *S. cerevisiae* GID complex are labeled in green. Conserved GID subunits currently not assigned in *S. cerevisiae* as *bona fide* GID proteins are colored in yellow. (D) Stably expressed HSS-Armc8 was isolated from HEK-293 cells, and after HA-peptide elution the presence of additional GID proteins in the immunoprecipitate was analyzed using the indicated antibodies. The asterisk (*) marks an unspecific strong band detected by the Twa1 antibody at approximately the same size in WCEs. (E) Transiently expressed HSS-Armc8 was immunoprecipitated from HEK-293 cells and probed by immunoblotting for the presence of the novel GID subunit YPEL5 and the RING protein MAEA. (F) Native GID-particles were purified from HEK-293 stably expressing HSS-tagged WDR26. The eluted complexes were visualized by SDS-PAGE and Sypro-Ruby staining (left panel) or subjected to an ubiquitination assay in the presence of Cdc34 and UbcH5b (right panel), with or without addition of E1 enzyme. The reaction was analyzed after the times indicated by SDS-PAGE followed by immunoblotting with the respective antibodies.

DOI: https://doi.org/10.7554/eLife.35528.002

The following source data and figure supplement are available for figure 1:

**Source data 1.** List of WDR26-interactors identified by AP-MS.
DOI: https://doi.org/10.7554/eLife.35528.004

**Source data 2.** List of Rmnd5a-interactors identified by AP-MS and SAINT analysis.
DOI: https://doi.org/10.7554/eLife.35528.005

**Source data 3.** List of Armc8-interactors identified by AP-MS and SAINT analysis.
DOI: https://doi.org/10.7554/eLife.35528.006

**Figure supplement 1.** Compositional analysis of the mammalian GID complex.
DOI: https://doi.org/10.7554/eLife.35528.003

network encompassing the entire set of homologous *S. cerevisiae* GID complex subunits (*Figure 1—figure supplement 1A*). This cluster additionally comprises the protein Mkln1, which was previously suggested to replace WDR26 in the human complex (*Figure 1—figure supplement 1A* and *Figure 1—source data 1*) (*Francis et al., 2013*; *Kobayashi et al., 2007*). By contrast, the CRL4 core components Ddb1 and Cullin4A/B were absent or not significantly enriched in WDR26 pull-downs (*Figure 1—source data 1*). Consistent with this observation, untagged WDR26 does not associate with the core CRL4 subunit Ddb1 in cells relative to the canonical CRL4 substrate adapter protein WDR23 (DCAF11) (*Figure 1—figure supplement 1B*). From these results, we conclude that WDR26 is not strongly linked to Cul4 proteins but rather a central subunit of the mammalian GID/CTLH complex.

While the *S. cerevisiae* GID complex is a characterized E3 ubiquitin ligase that targets gluconeogenic enzymes within a recently identified Pro/N-end rule pathway (*Chen et al., 2017*; *Santt et al., 2008*), data on the structural composition, enzymatic activity or biological function of the orthologous human CTLH complex is still missing. To fill this gap, we initiated a series of systematic AP-MS experiments of HA-2xStrep (HSS) -tagged versions of the GID subunits Rmnd5a and Armc8 to examine the global interactome of the complex in HEK-293 cells. Identification of high-confidence candidate interacting proteins (HCIPs) was based on semi-quantitative spectral counting by SAINT-scoring the datasets (*Choi et al., 2011*; *Mellacheruvu et al., 2013*). In accordance with the WDR26 dataset, the largest category of both Rmnd5a- and Armc8-HCIPs comprises the GID members RanBP9 (Gid1), c17ORF39 (Gid4), Armc8 (Gid5), Twa1 (Gid8), MAEA (Gid9) and Mkln1 (*Figure 1C*, *Figure 1—figure supplement 1C*, *Figure 1—source data 2*, *Figure 1—source data 3*). Furthermore, we identified Rmnd5b exclusively in Armc8 and WDR26 samples but not Rmnd5a, suggesting that the presence of the Rmnd5 paralogues in individual GID complexes is mutually exclusive. YPEL5 and RanBP10, a protein highly homologous to RanBP9, also cluster as GID subunits with high probability. To validate these proteomics data, we probed Armc8-, GID4- and Rmnd5a-immunoprecipitates for the presence of individual GID proteins using available antibodies against WDR26, YPEL5, Twa1, RanBP9 and MAEA (*Figure 1D and E*, *Figure 1—figure supplement 1D* and *Figure 3—figure supplement 1B*). Together, these experiments underscored that in cells the identified proteins assemble a stable and soluble heterodecameric functional unit, which we will from now on refer to as the hGID complex (*Figure 1—figure supplement 1E*). Interestingly, YPEL5 is homologous to the yeast protein Moh1, which was reported to associate with several GID subunits including Gid1, 2, 5, 7 and 8 (*Ho et al., 2002*). To test whether Moh1 is indeed a functional GID-subunit, we monitored degradation of the gluconeogenic enzyme Fbp1 after switching *moh1Δ* cells from a non-fermentable carbon source to glucose-containing media. However, Moh1-deficient cells degrade TAP-tagged Fbp1 with wild-type kinetics, whereas - as expected - Fbp1 levels remain high in cells lacking Gid4 (*Figure 1—figure supplement 1F*). We thus conclude that the yeast YPEL5, Moh1, is dispensable for catalytic GID activity.

Both Rmnd5 and MAEA harbor conserved C-terminal RING domains, which are predicted to bind upstream ubiquitination enzymes and typically confer ligase activity. To interrogate a potential E3 ubiquitin ligase function of the human complex, we purified native GID-immunocomplexes from HEK-293 cells and incubated them in the presence of ubiquitin, ATP and a mix of E1 and several E2 enzymes. As shown in *Figure 1F*, these reactions produced a pronounced ubiquitin smear, which is likely the result of autoubiquitination of one or several subunits in the absence of a cognate substrate. Thus, although residual contaminant E3 ubiquitin ligases in the sample preparation cannot be entirely excluded, these assays provide experimental evidence that the native human GID complex holds an intrinsic E3 ubiquitin ligase activity.

## Reconstitution of the human GID complex reveals a conserved architecture

With a molecular weight of ≥550 kDa, the mammalian GID complex potentially constitutes an exceptionally large E3 ubiquitin ligase of roughly half the size of the major cell-cycle regulator anaphase-promoting complex/cyclosome (APC/C) (*Zhang et al., 2013*). We sought to collect comprehensive topological and enzymatic information on the macromolecular GID complex by expressing all 10 full-length human GID subunits in Sf9 insect cells. To simultaneously identify potential stable GID subcomplexes within the putative heterodecamer, we applied a strategy of mixing various GID baculovirus combinations. After co-infection, the respective complexes were purified by means of single step

FLAG-pulldowns and peptide elution. This approach revealed that the conserved proteins RanBP9 and WDR26 bind the catalytic body of the E3 ligase, a trimeric complex composed of the two RING proteins Rmnd5, MAEA and the scaffolding protein Twa1 (*Figure 2A and E*). This stable pentameric assembly migrates as an individual peak in size-exclusion experiments, and is also catalytically active in autoubiquitination assays (*Figure 2B*, *Figure 2—figure supplement 1A* and data not shown). Densitometry quantification of Sypro Ruby-stained gels after gel filtration chromatography suggests a 2:1 molar ratio of Twa1 over the other subunits (*Figure 2C*). This is consistent with recent data where isolated untagged Twa1 was found as a constitutive dimer or oligomer (*Francis et al., 2017*). Indeed, the pentameric core GID complex shows an elution profile indicative of a large hydrodynamic radius or higher order assembly (*Figure 2—figure supplement 1A*). To analyze the absolute molecular mass of the GID pentamer in solution we subjected the subcomplex to size exclusion chromatography coupled to multi-angle static light scattering (SEC-MALS). SEC-MALS yielded two major peaks; the first peak, comprising all five GID proteins, with an average molecular mass of the 543 kDa, is consistent with the formation of a dimer (calculated molecular mass of the pentamer with two molecules of Twa1: ~285 kDa) (*Figure 2D*). The broad second peak encompasses protein assemblies ranging from ~155 kDa to 60 kDa which corresponds to the summarized mass of the catalytic Twa1::MAEA::Rmnd5a trimer (calculated molecular mass: 140 kDa) and dimerized Twa1 dissociated from the subcomplexes (calculated molecular mass of the Twa1: ~27 kDa) (*Figure 2B and D*). In yeast, deletion analysis to map Gid-Gid protein interactions identified the homologue of Twa1, Gid8, as the central hub connecting the RING dimer to the proteins Gid1 (RanBP9) and Gid7 (WDR26) (*Menssen et al., 2012*). Similarly, within the human complex, Twa1 is pivotal for efficient MAEA and Rmnd5a RING dimerization and downstream recruitment of the peripheral members of the protein complex in vitro (*Figure 2E* and data not shown). Together, the reconstitution data strongly predict cellular disintegration of the complex in the absence of the core protein Twa1. Indeed, when the complex was purified by immunoprecipitation from HEK-293 cells depleted for Twa1 using a pool of siRNAs, the otherwise stable GID network was largely disrupted (*Figure 2F*). Furthermore, step-wise in vitro reconstitution of the entire complex revealed that the pentameric core GID proteins form the platform strictly required for the downstream recruitment of the metazoan-specific protein Mkln1, YPEL5 and an independent unit formed by Armc8 and GID4 wherein GID4 only incorporates into the complex in the presence of Armc8 (*Figure 2—figure supplement 1B* and data not shown). Notably, both previously proposed yeast Gid7 homologues, WDR26 and Mkln1, coexist within a single GID complex (*Figure 2—figure supplement 1B*).

## Identification of Ube2H as cognate E2 enzyme for the human GID E3 ubiquitin ligase

The predominant mechanism of ubiquitin-transfer underlying E3 ligases are RING domain-containing proteins, which recruit and activate specific E2~Ub intermediates. With roughly 40 distinct E2 enzymes, various generic E2-E3 interactions can lead to productive ubiquitination events in vitro (*Deshaies and Joazeiro, 2009*). However, a growing body of evidence suggests that physiological E2-E3 pairs are critical for the rate and specificity of substrate ubiquitination (*Rape and Kirschner, 2004*; *Scott et al., 2016*). To investigate the molecular determinants of human GID E3 ligase activity, we reconstituted the full complex and subjected it to a series of autoubiquitination reactions screening a panel of distinct E2 enzymes (*Figure 2G*). Remarkably, when the hGID complex was mixed with Ube2H, a striking increase in autocatalytic hGID activity was observed, and ubiquitin was preferentially transferred to the RING protein MAEA (*Figure 2H*). No such activity was detected with any of the other tested E2 enzymes, including the promiscuous UbcH5 family and the SCF-specific E2 Cdc34 (*Figure 2—figure supplement 1C*). These data are supported by the presence of Ube2H peptides exclusively in Armc8- and WDR26-immunoprecipitates (*Figure 1—source data 1*, *Figure 1—source data 3*), and a weak but reproducible association of recombinant Ube2H and the hGID core complex in FLAG-based pulldown assays (*Figure 2—figure supplement 1D*). Notably, Ube2H displays 54% amino acid sequence identity to the yeast E2 Ubc8/Gid3, which is required to degrade gluconeogenic enzymes (*Kaiser et al., 1994*; *Schüle et al., 2000*). In summary, in vitro reconstitution analysis of the human GID complex highlights the remarkable organizational and catalytic conservation of this macromolecular E3 ubiquitin ligase across organisms.

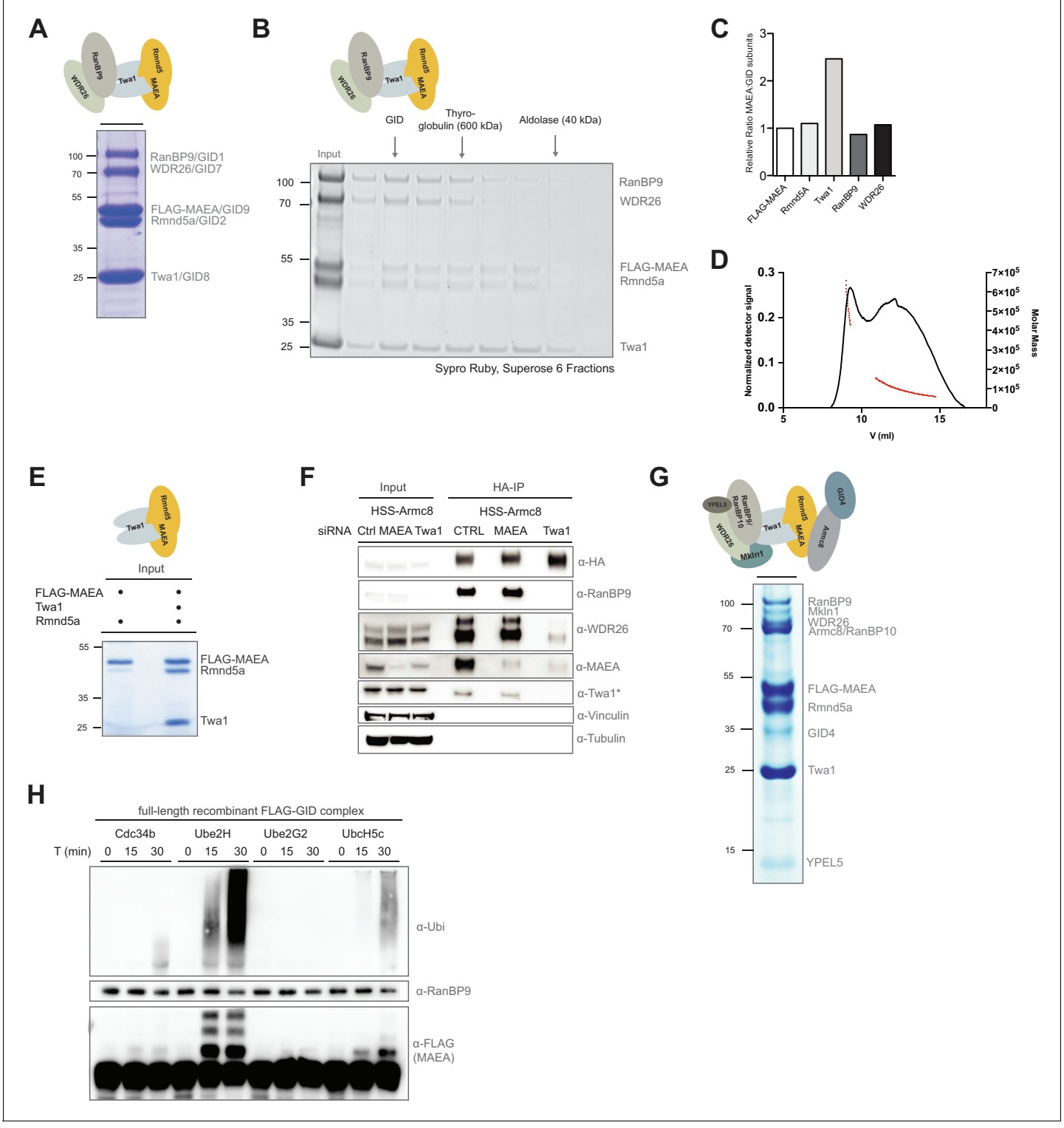

**Figure 2.** Reconstitution of the human GID E3 ligase complex. (**A**) Coomassie-stained SDS-PAGE showing a stable pentameric GID subcomplex expressed in insect cells and purified via the FLAG-tagged MAEA/GID9 subunit. All co-expressed proteins migrate at the predicted molecular weight. The molecular mass in kDa is indicated on the left. The deduced assembly of the GID subcomplex is schematically illustrated on top. (**B**) The pentameric GID complex was separated by size-exclusion chromatography and the fractions analyzed by Sypro-Ruby staining (left panel). The peak fraction is indicated (GID) relative to the size marker proteins. (**C**) The ratio of FLAG-tagged MAEA over co-eluting proteins in the peak fraction was quantified by densitometry analysis (right panel). Note that Twa1 is likely present as a dimer in the subcomplex. (**D**) SEC-MALS profile of the pentameric GID complex on a Superose six column. Over the center of the peak regions, the molecular weight of the molecules is indicated by the red line. (**E**) Co-
*Figure 2 continued on next page*

*Figure 2 continued*

expression of the RING proteins FLAG-MAEA and Rmnd5a in the presence or absence of Twa1. FLAG-peptide elutions of the individual purifications were analyzed side-by-side by SDS-PAGE/Coomassie staining. (F) Stably expressed HSS-Armc8 was isolated from HEK-293 cells treated for 72 hr with the indicated siRNAs. After HA-peptide elution the presence of several GID proteins was analyzed by immunoblotting. Note that in the absence of Twa1 the complex disintegrates. The asterisk (*) marks an unspecific strong band detected by the Twa1 antibody. (G) In vitro reconstitution of the human FLAG-tagged GID complex comprising all 10 full-length subunits from Sf9 cells. Single-step FLAG-purification analyzed by SDS-PAGE and subsequent Coomassie staining. The organization of the hGID is schematically illustrated on top. (H) Autoubiquitination capacity of the FLAG-tagged recombinant GID complex was tested in the presence of E1, Ubiquitin (Ubi) and $Mg^{2+}$•ATP and either Cdc34, Ube2H, Ube2G2 or UbcH5c. Over the indicated time-course, Ube2H most efficiently promoted ubiquitin chain assembly and specific ubiquitination of the FLAG-tagged GID subunit MAEA, as visualized by means of immunoblotting using anti-Ubiquitin and anti-FLAG-antibodies. Immunoblotting of RanBP9 controls the presence of purified GID complexes.

DOI: https://doi.org/10.7554/eLife.35528.007

The following figure supplement is available for figure 2:

**Figure supplement 1.** In vitro reconstitution of the human GID E3 ligase complex.

DOI: https://doi.org/10.7554/eLife.35528.008

## GID E3 ligase activity is critical for cell proliferation

Since WDR26 is important for normal cell growth, we interrogated the role of the mammalian GID E3 ligase as a complex by generating a second GID CRISPR knockout (KO) RPE cell line with selective deletion of the RING protein MAEA. Like WDR26-KOs, MAEA-deficient cells are characterized by a remarkable decrease in their proliferative capacity compared to cells treated with control gRNA in MTT and clonogenic assays (*Figure 3A and C*, data not shown). As expected, deletion of the RING protein renders the ligase inactive and results in altered GID complex assembly and subunit stabilization, suggesting an autoregulatory mechanism frequently observed for E3 ligases (*Figure 3B*, *Figure 3—figure supplement 1B*) (*Metzger et al., 2012*). Immunoblotting of cell extracts revealed that the mitotic marker phospho-Histone H3 (Ser10) was absent in MAEA- and WDR26-KO cells (*Figure 3D*, *Figure 3—figure supplement 1D*). Furthermore, the decrease in cell proliferation was not due to an increment in apoptotic cell death (*Figure 3—figure supplement 1C*), but is manifested by a pronounced downregulation of both the levels and phosphorylation of the retinoblastoma protein (Rb) and of several other cell cycle markers including Cyclin A and to a lower extent Cyclin D1 (*Figure 3D*, *Figure 3—figure supplement 1C*). In agreement with these observations, phosphorylation of nuclear pRB by CDKs is known to be crucial for cell cycle entry to unleash the activities of growth-promoting transcription factors such as the E2F family (*Brown et al., 1999*; *Rizzolio et al., 2012*). Curiously, cells lacking hGID activity adapt within days of culturing, restoring normal cell proliferation (*Figure 3C*), accompanied by re-expression of all cell cycle markers that are nearly indistinguishable to control cells (*Figure 3D*). Thus, we speculate that GID-KO cells rapidly acquire suppressor mutations and/or activate a compensatory pro-proliferative transcriptional program that overrides the loss of hGID-activity.

## The mammalian GID complex is dispensable for degrading gluconeogenic enzymes

To identify the role of hGID in promoting cell proliferation in mammalian cells, we first tested whether like in yeast the E3-ligase complex targets excess gluconeogenic enzymes upon a metabolic switch. Surprisingly, most human gluconeogenic enzymes do not harbor N-terminal proline residues, which was recently postulated as main recognition motif for the yeast GID E3-ligase (*Chen et al., 2017*). The relative abundance of the gluconeogenic enzyme FBP1 was first assayed in RPE and HEK-293 cells lacking GID activity under normal growth condition. Indeed, we found that FBP1 levels remained unchanged in MAEA-KO cells, in contrast to the autoregulated hGID subunit YPEL5 (*Figure 3E*, data not shown). To corroborate this result, we next analyzed the rate-limiting gluconeogenic enzymes FBP1 and PCK1 levels in HEK-293 cells and primary mouse hepatocytes treated with control siRNA or siRNA oligos targeting all GID subunits undergoing a metabolic switch from gluconeogenesis to glycolysis (*Figure 3—figure supplement 1D*). To this end, HEK-293 cells and freshly isolated primary hepatocytes were cultured in gluconeogenic conditions for 6 hr in glucose-free medium supplemented with pyruvate, lactate and forskolin. FBP1 and PCK1 protein levels were assayed for up to 5 hr after switching to glycolytic conditions in glucose-rich medium (*Figure 3F,G*).

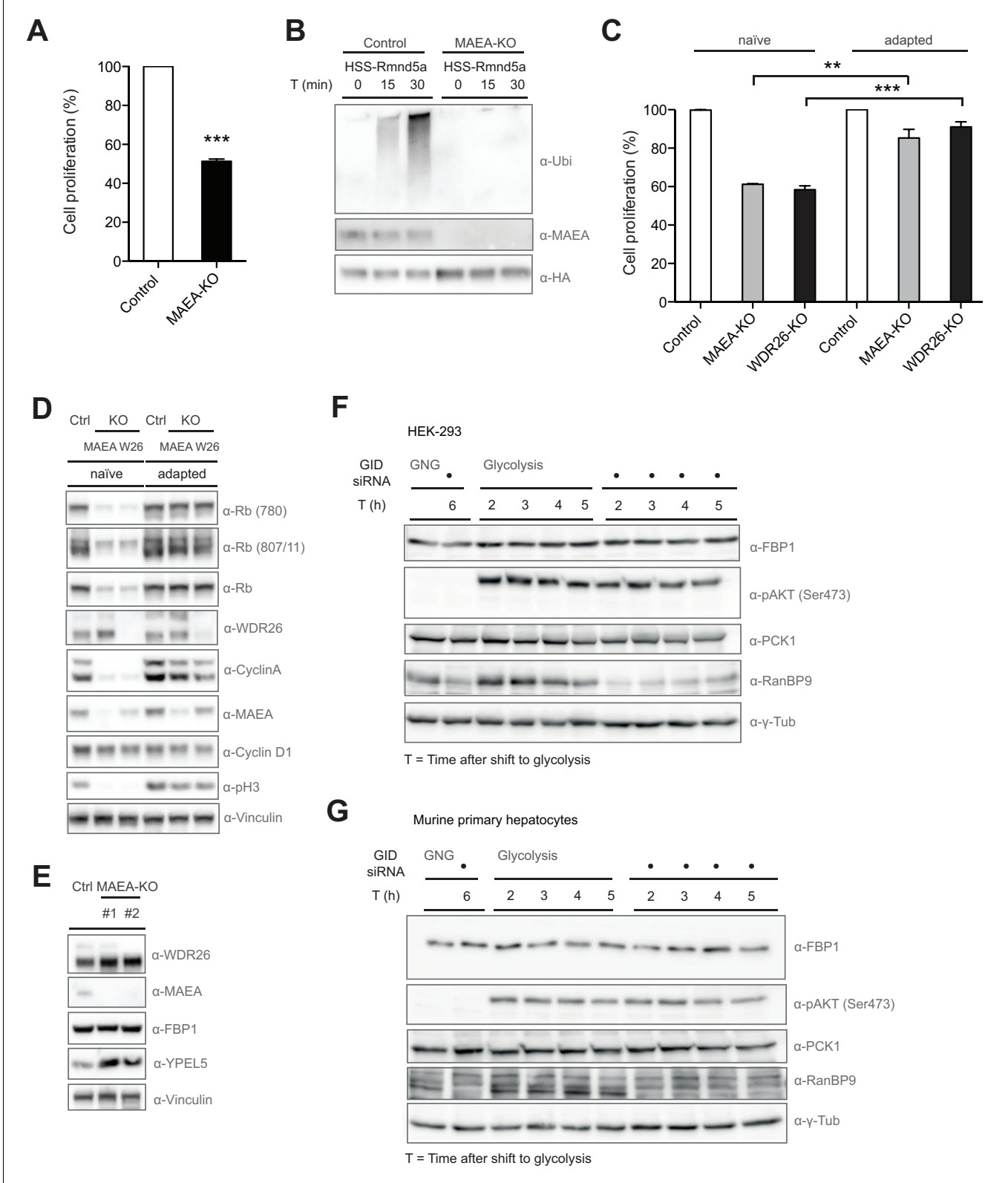

**Figure 3.** Compromised GID activity results in an intermittent cellular growth defect. (**A**) Cell proliferation of RPE control and MAEA-KO cells was quantified with MTT assays between days 6 and 9 after lentiviral transfection. Data are shown as mean of triplicates and % change in signal relative to control gRNA-treated cells ± SD, n = 3, ***p≤0.0006. (**B**) In vitro autoubiquitination assay of native GID particles isolated via the stably expressed HSS-tagged RING protein Rmnd5a from HEK-293 control or MAEA-KO cells in the presence of Ube2H. Deletion of MAEA results in complete loss of the
*Figure 3 continued on next page*

*Figure 3 continued*

catalytic activity of the GID complex as shown by immunoblotting using an antibody directed against Ubiquitin. (**C**) Cell proliferation of RPE MAEA-KO and WDR26-KO cells was measured by MTT assays between days 6–9 (naïve) and 15–17 (adapted) after lentiviral transfection. Data are shown as mean of triplicates and % change in signal relative to control gRNA-treated cells ± SD, n =≥2, **p≤0.0016, ***p≤0.0007. (**D**) Cell extracts prepared from naïve and adapted RPE MAEA-KO and WDR26-KO cells were analyzed for cell cycle and growth pathway markers by immunoblotting. Note that the adapted GID-KO cells overcome the proliferation defect as judged by the presence of phosphorylated Histone H3, Rb (S780 and S807/811) and restored levels of the cell cycle markers Cyclin A and Cyclin D1. (**E**) Untreated (naïve) control and MAEA-KO RPE cells express comparable FBP1 protein levels in contrast to the autoregulated and stabilized GID subunit YPEL5. (**F** and **G**) The murine GID complex is dispensable for degradation of the gluconeogenic enzymes FBP1 and PCK1 in HEK-293 and primary murine hepatocytes. Cells were starved after control and GID siRNA treatment before switching them back to glycolytic conditions. Samples for immunoblot analysis were taken at the indicated time points and probed for FBP1 and PCK1 protein levels (n = 2). The metabolic switch in these cells was controlled by phosphorylation of Akt (Ser473) and the efficiency of GID complex depletion was probed using an antibody directed against the subunit RanBP9 of the complex.

DOI: https://doi.org/10.7554/eLife.35528.009

The following figure supplement is available for figure 3:

**Figure supplement 1.** GID activity is required for cell growth in RPE cells.

DOI: https://doi.org/10.7554/eLife.35528.010

To confirm that these cells indeed changed their metabolic program, we also followed insulin-stimulated phosphorylation of Akt at Ser$^{473}$ (*Puigserver et al., 2003*). Unlike yFbp1 and yPck1, which are rapidly degraded in the presence of glucose in a GID-dependent manner (*Chen et al., 2017*), mFBP1 and mPCK1 protein levels remained relatively stable over the time-course and unresponsive to GID knockdown. Moreover, FBP1 mRNA levels drop after shifting cells from a fasted to a fed state irrespective of the presence of the GID complex in primary hepatocytes (*Figure 3—figure supplement 1E*). Additionally, knockdown of GID did not alter glucose production or glucose levels when primary hepatocytes were shifted from gluconeogenesis to glycolysis as would be expected if the GID complex takes on the same function as in yeast (*Figure 3—figure supplement 1F,G*). Collectively, these results demonstrate that neither mammalian FBP1 nor PCK1 are targets of the GID complex, indicating a substantial functional divergence of the GID E3 ligase between yeast and mammals.

## Semi-quantitative proteomics identifies novel and regulated GID interactions

To isolate novel physiological substrates of the hGID E3 ligase, we carried out a second round of HSS-Rmnd5a AP-MS experiments in the presence of the 26S proteasome inhibitor MG132 (*Figure 4A* and *Figure 4—source data 1*). Interacting proteins with a high-confidence score were clustered into the following categories: (1) proteins that stably interact with the GID complex irrespective of MG132-treatment including the GID members themselves, the mitochondrial protease HTRA2 and the poorly characterized protein ZMYND19; (2) proteins that were only detected upon MG132-induced proteotoxic stress such as multiple proteasome and several subunits of the protein quality control BAG6/GET complex; and (3) low abundant but recurring proteins that are substantially enriched in MG132-treated samples such as the transcription factor Hbp1 (*Figure 4B and C*). To corroborate the robustness of the data, we performed immunoprecipitation experiments using HEK-293 cell lines stably expressing either HSS-tagged Armc8 or Rmnd5a from a doxycycline-inducible promoter. While the GET complex subunit BAG6 was only weakly detectable in GID-immuno-complexes, the conserved HTRA2 protease emerged as a strong interactor (*Figure 4D* and *Figure 4—figure supplement 1B*). This serine protease was initially proposed to function as a pro-apoptotic molecule inducing cell death under adverse growth conditions, but was recently found to generally contribute to cellular homeostasis (*Kuninaka et al., 2007*). Consistent with the MS-data, we did not observe a significant change in HTRA2 levels in MAEA or WDR26 deleted HEK-293 or HeLa Kyoto cells (*Figure 4—figure supplement 1A*), implying that HTRA2 is unlikely a proteolytic GID substrate. Next, we focused our analysis on the high mobility group (HMG) box-containing protein 1 (Hbp1), which is strongly enriched in MG132-treated samples (*Figure 4C*), indicative of a UPS-regulated interaction. Hbp1 was previously reported to regulate the expression of important cell cycle regulators such as N-Myc, Cyclin D1, p16 and p21, thereby triggering a slowdown of cell growth and premature senescence (*Li et al., 2010*; *Sampson et al., 2001*; *Tevosian et al., 1997*;

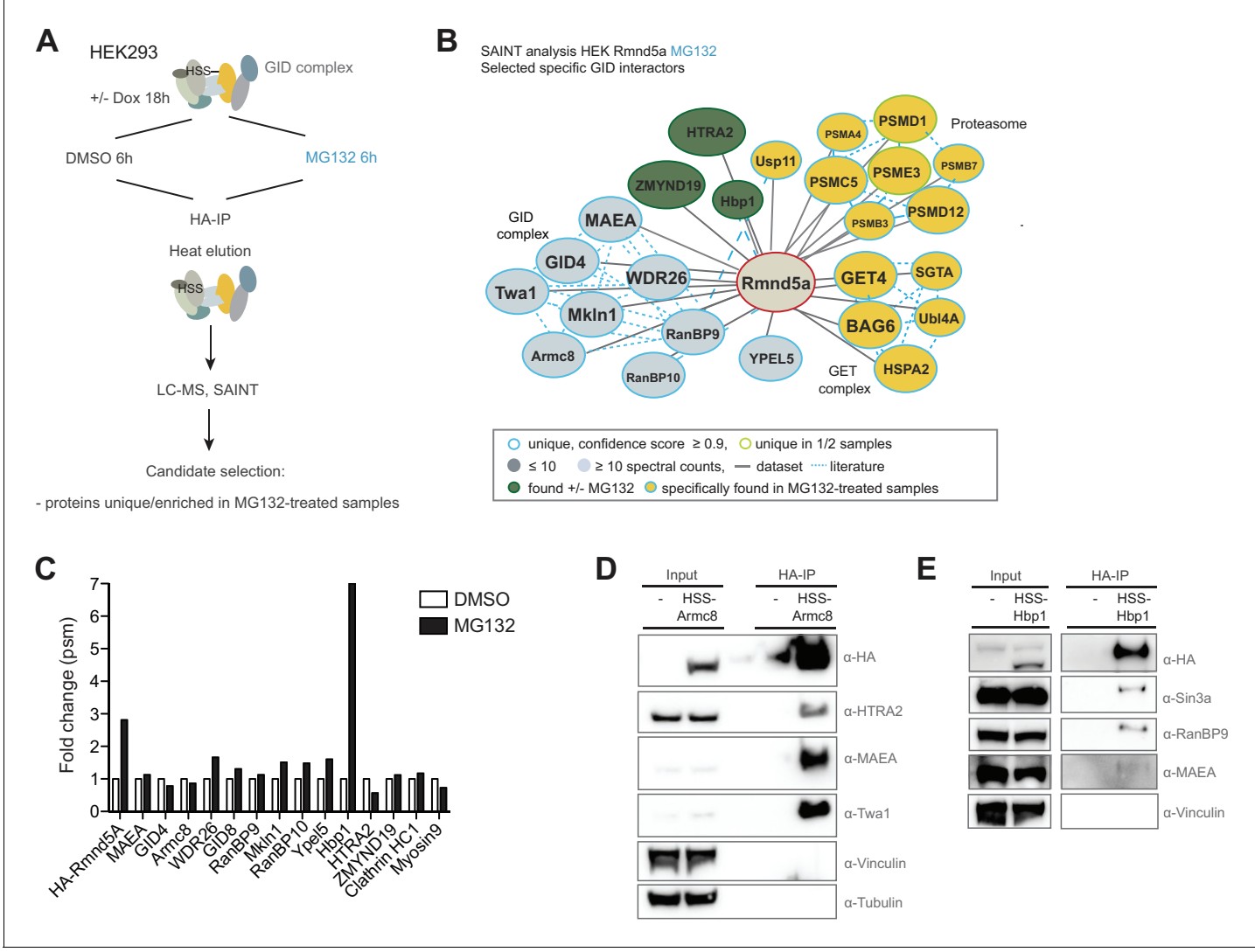

**Figure 4.** Semi-quantitative proteomics identifies novel stable and regulated GID-interacting proteins. (**A**) Scheme of the applied AP-MS workflow to identify MG132-enriched Rmnd5a-HCIPs in HEK-293 cells. (**B**) SAINT-network of Rmnd5a-HCIPs in cells treated with MG132 for 6 hr (confidence score ≥0.9, FC ≥ 4, n = 2). The GID subunits are labeled in light blue and remain stably associated in the presence of MG132. Constitutive HCIPs that were recovered in control and MG132-treated samples are colored in green. Proteins and protein networks that are specifically associated with the GID complex in the presence of MG132 are highlighted in yellow. (**C**) Quantification of total peptide spectral matches (psm) measured for Rmnd5a-HCIPs in DMSO control vs. MG132-treated samples. Note that the transcription factor Hbp1 is specifically enriched upon proteasome inhibition. (**D**) Co-immunoprecipitation experiment (HA-IP) using HEK-293 cells stably expressing HSS-Armc8 demonstrates specific interaction of the mitochondrial protease HTRA2 with the GID complex. (**E**) Immunoprecipitation and subsequent immunoblot analysis of HEK-293 cells expressing a doxycycline-inducible HSS-tagged construct of Hbp1. Note that Hbp1 not only binds its corepressor protein Sin3a but also multiple subunits of the GID complex.
DOI: https://doi.org/10.7554/eLife.35528.011

The following source data and figure supplement are available for figure 4:

**Source data 1.** List of Rmnd5a-interactors in the presence of MG132 identified by AP-MS and SAINT analysis.
DOI: https://doi.org/10.7554/eLife.35528.013

**Source data 2.** List of Hbp1-interactors identified by AP-MS and SAINT analysis.
DOI: https://doi.org/10.7554/eLife.35528.014

**Figure supplement 1.** The transcription factor Hbp1 is a novel reciprocal GID-interacting protein.
DOI: https://doi.org/10.7554/eLife.35528.012

*Wang et al., 2012*). Since endogenous Hbp1 levels are low in cycling cells, we generated inducible HEK-293, HeLa Kyoto and U2OS cell lines expressing HSS-tagged Hbp1 to substantiate a reciprocal GID-Hbp1 interaction. Indeed, Hbp1-immunoprecipitates comprise its known binding partner and transcriptional corepressor Sin3A and the GID subunits RanBP9 and MAEA (*Figure 4E*) (*Swanson et al., 2004*). Moreover, AP-MS followed by SAINT analysis recovered two large functional clusters, encompassing multiple subunits of the Sin3A corepressor complex and the entire set of human GID subunits (*Figure 4—figure supplement 1C* and *Figure 4—source data 2*). Since the GID complex predominantly accumulates in the nucleus (*Figure 3—figure supplement 1A*), together these results support the hypothesis that the GID complex might target nuclear Hbp1 for proteasomal degradation.

## Hbp1 protein levels are directly controlled by the GID E3 ligase

To investigate whether Hbp1 might be a true ubiquitination substrate of the GID E3 ligase, Hbp1 proteins levels were examined upon downregulation of different GID subunits. To minimize off-target effects, siRNA pools were employed to individually deplete WDR26, Rmnd5a, Twa1 and MAEA in Hela Kyoto, HEK-293, RPE and untransformed BJ cells. Remarkably, Hbp1 levels increased approximately 3-fold across all cell types tested (*Figure 5A and B*, *Figure 5—figure supplement 1A and B*). In contrast, Hbp1 mRNA levels quantified by qRT-PCR in GID-depleted cells are comparable to controls (*Figure 5C*), excluding the possibility that elevated Hbp1 levels are indirectly caused by upregulated Hbp1 transcription. Complementation assays with inducible expression of a RNAi-resistant version of WDR26 confirmed that Hbp1 stabilization is a direct consequence of crippled GID activity (*Figure 5D*). Moreover, Hbp1 levels were also upregulated in both WDR26-KO and MAEA-KO cells accompanied with a slightly reduced expression of Cyclin D1, a presumed Hbp1 transcriptional target (*Figure 5E*) (*Sampson et al., 2001*). Consistent with this observation, we found total Cyclin D1 mRNA partially repressed in WDR26-KO cells, correlating in intensity with ectopic overexpression of Hbp1 (*Figure 5—figure supplement 1C and D*). To investigate whether hGID E3 ligase activity alters Hbp1 protein stability, we treated control and WDR26-depleted cells with cycloheximide (CHX) and quantified Hbp1 levels at different time intervals. Notably, the lack of WDR26 strongly reduced the rate of Hbp1 degradation (*Figure 5F*), suggesting that the complex is required for Hbp1 degradation in vivo. Intriguingly, we noticed that the human GID substrate Hbp1 contains an alternative start site just 10 amino acids downstream of its canonical sequence that conforms to a perfect yeast GID Pro/N-end rule recognition motif (*Chen et al., 2017*). To test a potential functional relevance of these residues for Hbp1 degradation, we mutated the critical proline to a glycine (P11G). However, the Hbp1-P11G mutant was degraded with kinetics similar to the wild-type protein (*Figure 5—figure supplement 1G*), arguing against a conserved Pro/N-end rule pathway consensus targeting sequence for the case of Hbp1.

To examine whether Hbp1 can be directly modified by the GID E3 ligase, we expressed and purified C-terminally tagged Hbp1 from Sf9 insect cells and performed in vitro ubiquitination assays in the presence of recombinant GID complex and Ube2H (*Figure 5—figure supplement 1F*). Indeed, Hbp1 was specifically ubiquitinated by hGID but not by the control E3 ligase CRL4$^{CRBN}$ (*Figure 5F*). To dissect the relative substrate specificity of the GID complex in vitro, we also tested the catalytic activity of the complex towards the yeast specific GID target yFbp1. However, although we observed GID autoubiquitination, we were unable to detect ubiquitinated yFbp1 in these experiments (*Figure 5—figure supplement 1H*). Together, these data establish that Hbp1 is a proteolytic substrate of the human GID E3 ligase complex, and indicate that degradation of Hbp1 might be required for efficient cell proliferation.

## Discussion

Here, we combined systematic proteomics, biochemical reconstitution and cellular approaches to unravel the architecture, enzymatic features and functional roles of the metazoan counterpart to the *S. cerevisiae* E3 ligase GID. Our results illustrate its evolutionary diversification, identify a *bonafide* substrate for its hitherto orphan enzymatic activity and most importantly underscore that GID-activity is critical for cell proliferation in mammals. Together, these findings not only highlight the physiological relevance and regulation of this macromolecular machine, but also serve as a basis for acquiring further molecular insights into the mechanisms of GID substrate selection and ubiquitin transfer.

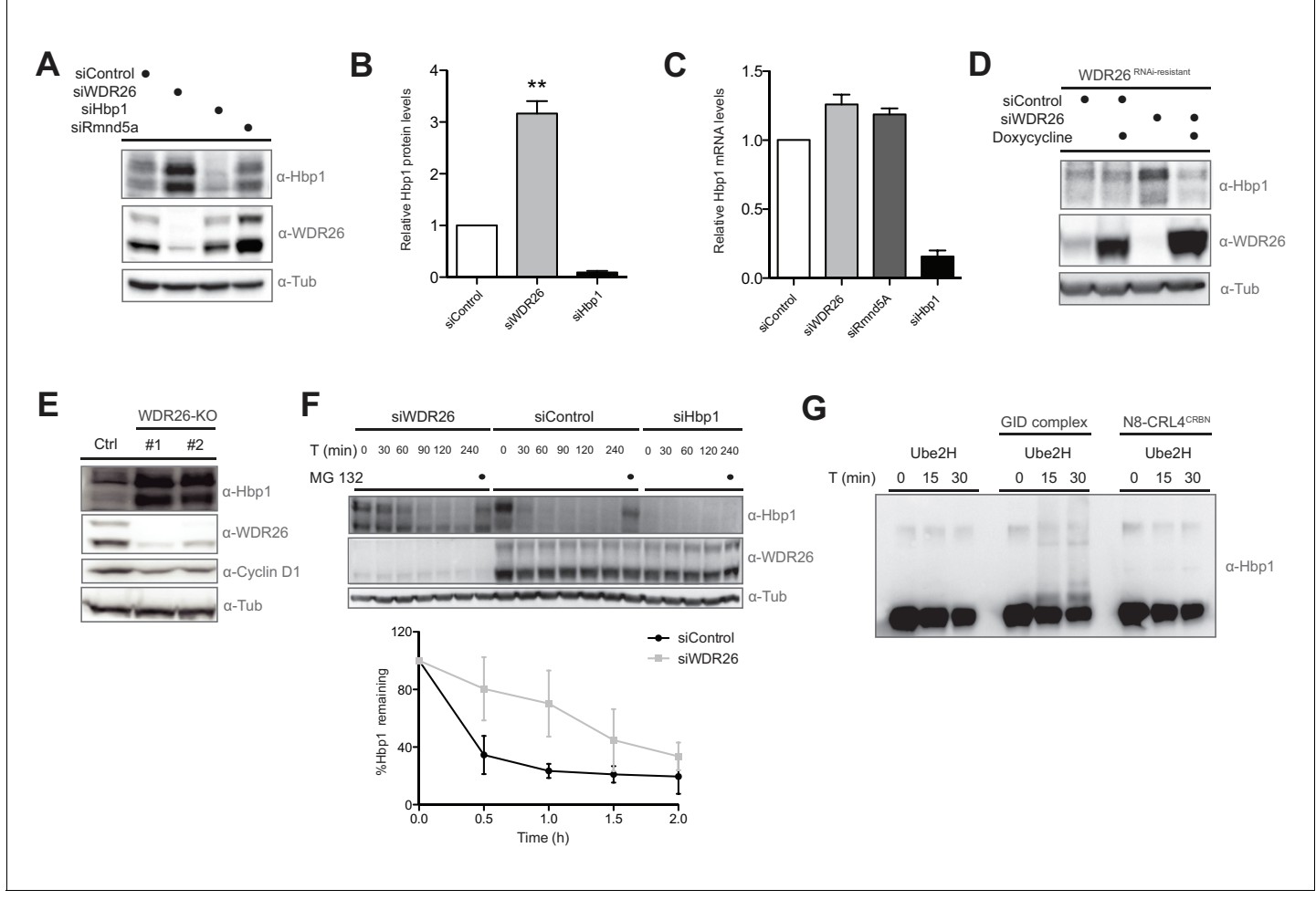

**Figure 5.** The transcription factor Hbp1 is a proteolytic ubiquitination target of the human GID E3 ligase. (**A**) Depletion of WDR26 and Rmnd5a with pools of siRNAs for 72 hr increase the protein levels of the transcription factor Hbp1 in HeLa Kyoto cells. (**B**) Corresponding quantification of the relative proteins levels of Hbp1 and WDR26 upon downregulation of WDR26 compared to control siRNA-treated samples. Average data ± SEM, n ≥ 3 independent experiments, **p≤0.006. (**C**) qRT-PCR analysis of total Hbp1 mRNA levels in HeLa Kyoto cells upon RNAi-mediated depletion of WDR26, Rmnd5a and Hbp1. Data are shown as mean fold change relative to control siRNA-treated cells ± SD, n = 3. (**D**) Immunoblot analysis of HeLa Kyoto cells stably expressing from a doxycycline-inducible promoter an untagged WDR26$^{RNAi-resist}$ construct. As indicated, the cells were treated for 72 hr with control siRNA or siRNA oligos depleting endogenous WDR26. (**E**) HeLa Kyoto WDR26-KO cells generated with two different sets of gRNAs were immunoblotted for protein levels of WDR26, Hbp1 and Cyclin D1. (**F**) The half-life of endogenous Hbp1 was determined in HeLa Kyoto cells treated with siRNA control or RNAi-depleted of WDR26 or HBP1 for 72 hr. Protein translation was blocked by the addition of 50 μg/μl cycloheximide, and Hbp1 levels were analyzed at the indicated time points after drug treatment. (**G**) Ubiquitination of Hbp1-FLAG in vitro. Purified Hbp1 was incubated with ubiquitin, E1, Ube2H and the full-length recombinant GID complex or control neddylated CRL4$^{CRBN}$ at 37°C for the indicated time (min). Samples were analyzed by immunoblotting with an antibody directed against Hbp1.

DOI: https://doi.org/10.7554/eLife.35528.015

The following figure supplement is available for figure 5:

**Figure supplement 1.** The transcription factor Hbp1 is regulated by the human GID complex.

DOI: https://doi.org/10.7554/eLife.35528.016

## Architectural and enzymatic conservation of the core human GID/CTLH E3 ligase complex

It has been previously suggested that orthologs of the budding yeast GID complex are present in most phyla (*Francis et al., 2013*). However, until now it was unclear if these complexes entail organism-specific variations in their composition and enzymology. By merging systematic proteomics with in vitro reconstitution, we discovered remarkable parallels between yeast and human GID complexes. First, the catalytic core is compositely formed by the RING proteins MAEA and either

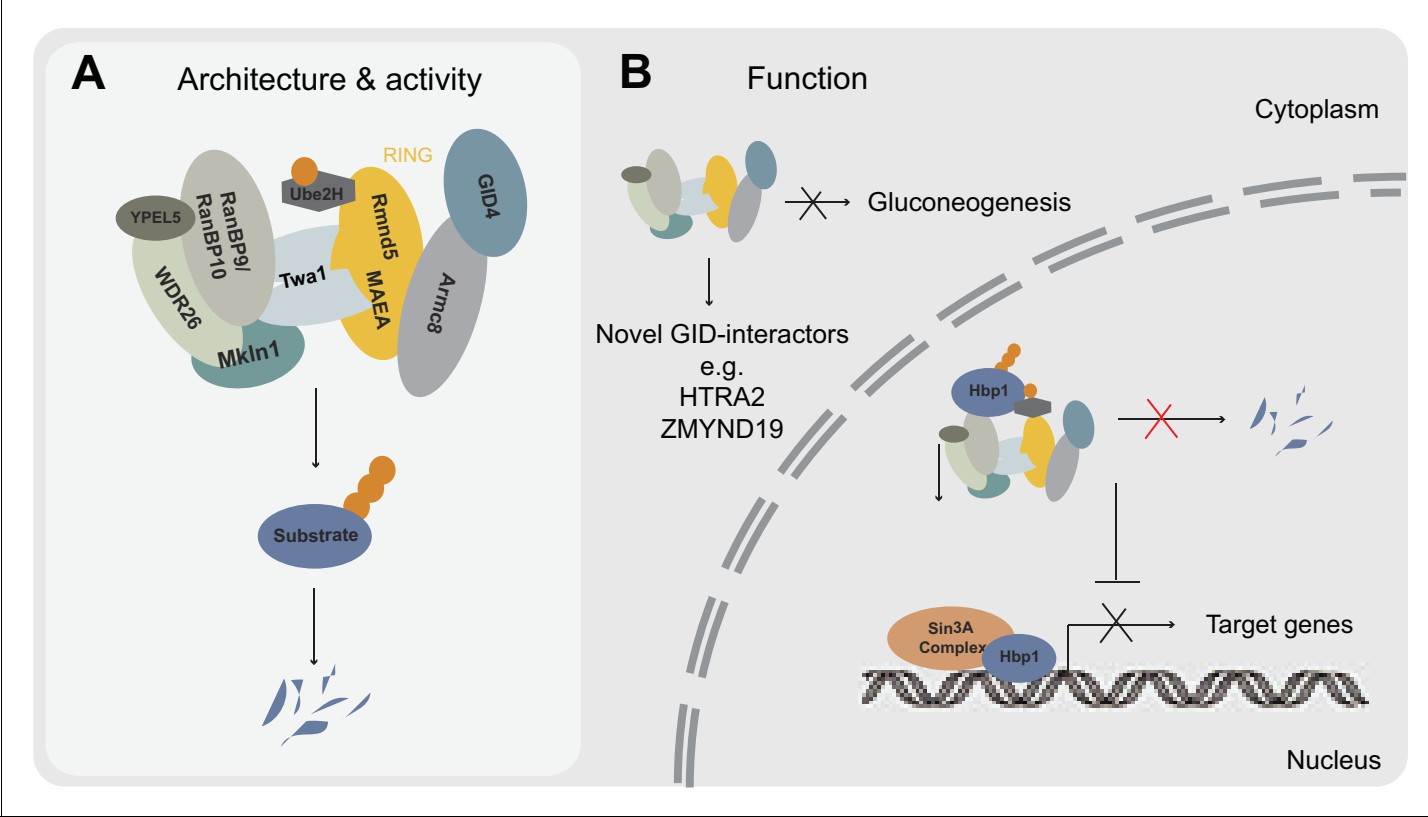

**Figure 6.** Model illustrating newly identified and potential GID E3 ligase functions in human cells. (**A**) Schematic representation of the mammalian GID complex summarizing its subunit composition, potential architecture and major catalytic features. In brief, the multi-subunit E3 ligase complex is formed around a stable catalytic core formed by the RING proteins MAEA and Rmnd5. The RING proteins MAEA and Rmnd5 are strictly required for normal complex formation and catalysis. Twa1 is a dimer that is critical for association of both RING proteins and downstream complex assembly. Armc8 is required for recruitment of GID4. The E2 enzyme Ube2H cooperates with the GID complex in substrate ubiquitination. (**B**) Based on our results we suggest that the GID complex is a conserved and ubiquitously expressed E3 ligase that resides both in the nucleus and cytoplasm, and regulates cell proliferation in human cells. While it remains to be investigated if and how the activity of the complex is spatially and/or temporally regulated, we speculate that the more peripheral subunits such as WDR26, RanBP9, Mkln1 and possibly GID4 may play important roles in anchoring the complex at specific subcellular locations and/or provide distinct docking sites for substrates. This might in part help to explain the plethora of interacting proteins reported for RanBP9 (*Salemi et al., 2017*). We were neither able to identify an obvious role for the metazoan GID complex in targeting gluconeogenic enzymes for proteasomal degradation nor find direct evidence for a potentially conserved function in a Pro/N-end rule pathway. A series of AP-MS experiments however, isolated novel GID-interacting proteins including the transcription factor and negative regulator of several pro-proliferative genes Hbp1. Hpb1 protein turnover is regulated by the GID E3 ligase and biochemical evidence suggests that Hbp1 is directly ubiquitinated by the complex, which might contribute to the intermittent proliferation phenotype observed for GID-KO cell lines.

DOI: https://doi.org/10.7554/eLife.35528.017

Rmnd5a or its paralog Rmnd5b (*Figure 6A*). Interestingly, RING dimerization is a functional requirement of several non-catalytic RING E3 ligases to stabilize the E2 ~ubiquitin thioester bond for ubiquitin transfer. Dimerization can be either mediated by the RING domain alone or facilitated by flanking helical regions (*Berndsen and Wolberger, 2014*). Interestingly, for the GID complex, we observed that optimal RING association is dependent on the small scaffolding protein Twa1. Our data additionally suggest that Twa1 itself dimerizes to form a central hub for the downstream recruitment of the core and peripheral members of the protein complex. A similar dependency on Gid8/Twa1 for complex assembly has also been reported for the yeast complex (*Menssen et al., 2012*). Together, Twa1, the RING proteins, RanBP9, WDR26 and Mkln1 form a stable hexameric complex in vitro. Furthermore, Armc8 incorporation is imperative for subsequent GID4 binding. In yeast, Gid4 association is tightly regulated and the protein acts both as substrate adapter for gluconeogenic enzymes as well as an activator of the E3 ligase activity (*Chen et al., 2017*; *Santt et al., 2008*). Mechanistically, Gid4 may induce a conformational change that allows for more efficient E2

binding and/or optimal positioning of the catalytic core towards the substrate. However, our data suggest that at least in vitro the human GID complex without the Armc8/GID4-module efficiently cooperates with Ube2H in autoubiquitination assays. Structural analysis will be required to establish a complete map of intra- and intermolecular contact sites and to understand the underlying topological integrity and enzymatic activity of the hGID complex. For example, it remains open if the highly homologous proteins RanBP9 and RanBP10 co-reside in individual complexes as suggested by our proteomics data. Moreover, it will be important to dissect the structural determinants that explain the Ube2H specificity and activation of ubiquitin-transfer by the RING proteins Rmnd5 and MAEA.

## Functional heterogeneity of eukaryotic GID complexes

In yeast, the GID complex is activated when the carbohydrate metabolism is reversed and cells rapidly switch from gluconeogenesis to glycolysis. In this process, GID activity balances the excess gluconeogenic enzymes. Surprisingly, despite the structural evolutionary conservation, the levels of the rate-limiting gluconeogenic enzymes FBP1 and PCK1 are unresponsive to GID depletion in mammalian cells. However, previous findings identified a specific role for the mammalian N-end rule enzyme UBR5 in degrading the gluconeogenic enzyme PCK1 under high-glucose conditions (*Jiang et al., 2011*) and a MAGE-TRIM28 E3 ligase complex targeting FBP1 for proteolytic degradation in a hepatocellular cancer background (*Jin et al., 2017*). Taken together, despite the UPS playing an important part in fine-tuning the concentration of gluconeogenic enzymes in metazoans, the mechanisms and enzymes involved appear fundamentally different. Compared to unicellular organisms, higher eukaryotes are highly specialized within tissues and organs in their internal biochemistry. Gluconeogenesis is largely restricted to hepatocytes and only activated upon extended periods of fasting. However, the GID complex is ubiquitously expressed, pointing towards an extended substrate repertoire, possibly depending on the prevailing cellular milieu and/or metabolic programs of different cell types. Interestingly, we found that hGID complexes interact with the mitochondrial protease HTRA2, suggesting that the GID E3-ligase may play a more general role in metabolism and/or protein quality control under normal growth or specific stress conditions.

Notably, a shared feature of all identified yeast GID substrates to date are proline residues in the first or second position after their start methionine and the presence of this amino acid is critical for GID-mediated degradation (*Menssen et al., 2012*). Recently it was postulated that the yeast GID complex operates within a novel Pro/N-end rule pathway (*Chen et al., 2017*). Indeed, the substrate-binding pocket of yeast and human Gid4 is highly conserved and human GID4 is able to recognize and bind various proline-containing N-degrons (Dong et al., 2018). Our complete list of Armc8 interactors include proteins, such as the RNA-helicase proteins DDX50 and DDX21, that conform to the consensus sequence of the mammalian Pro/N-end rule protein quality control branch (*Figure 1— source data 3* and data not shown) (*Chen et al., 2017*; *Dong et al., 2018*). However, the hGID-substrate Hbp1 identified here does not possess a proline-containing N-degron. Since our AP-MS data suggest that GID4 is more loosely associated with the complex than other subunits (*Figure 1— source data 2*, *Figure 1—source data 3*), we speculate that different mechanisms for regulated substrate recruitment may exist for mammalian GID E3 ligase complexes.

## The GID complex controls cell proliferation and targets the transcription factor Hbp1

Analyses of RanBP9-KO mice revealed that this GID subunit is critical for early embryonic development, but some pups survive and are significantly smaller than their wild-type littermates (*Palavicini et al., 2013*; *Puverel et al., 2011*). Available evidence suggests that individual deletions of critical GID subunits, such as the RING protein MAEA and the structural integrity protein Twa1, result in proliferation defects in several cell lines (*Hart et al., 2015*). This cell cycle defect is characterized by loss of mitotic markers and significantly reduced expression of critical cell cycle markers such as Cyclin A and pRb, characteristic for cells exiting into a quiescence state. Interestingly however, this strong cell cycle defect is only transient, and after a few days, individual cells resume normal proliferation rates and are morphologically and biochemically indistinguishable from control cells. While at present we do not know the molecular mechanisms underlying this suppression, we conclude that the hGID complex is not essential for cell proliferation per se, but negatively regulates one or several cell cycle inhibitors, possibly involved in cell cycle exit. Interestingly, our targeted AP-

MS experiments identified the transcription factor Hbp1 as a proteolytic target of the E3 ligase in various human cell lines (*Figure 6B*). Hbp1 was proposed to function as a tumor suppressor and has been implicated in regulating cell cycle exit. Indeed, cells overexpressing Hbp1 were previously shown to enter a quiescent-like $G_0$ state (*Shih et al., 2001*). Molecularly, Hbp1 executes dual transcriptional functions by directly binding high-affinity elements on negatively regulated target genes, including N-Myc and DNA methyltransferase 1 (DNMT1), but also transcriptionally activates genes, such as p16, p21 and histone H1 (*Pan et al., 2013*). Interestingly, our proteomic data suggest that the majority of Hbp1 is tightly embedded into a multi-subunit Sin3a corepressor complex. However, in sharp contrast to Hbp1, Sin3a-related proteins are excluded from the GID-interactome. Therefore, it is conceivable that continuous GID-mediated proteolytic clearance of Hbp1 in cycling cells is installed to avert the assembly of a repressor complex that controls the expression of a set of anti-proliferative genes (*Figure 6B*). While attractive, we were unfortunately unable to directly test this hypothesis, as GID-depleted cells rapidly acquired suppressor mutations. Moreover, while immuno-blotting of adapted GID-KO cells might suggest partial normalization of Hbp1 protein abundance, the effect is not robust enough to account for the phenotype reversal (*Figure 5—figure supplement 1E*). Given the high expression of the predominantly nuclear E3 ligase, it is not unlikely that GID-deficient cells have to cope with multiple deficiencies resulting from deregulated nuclear proteins. Deconvolution of the global molecular signature of the GID complex using transcriptomics in a side-by-side comparison with cells of differential GID-activity states should clarify whether transcriptional plasticity is an important mechanism to promote acquired resistance of specific cell types to loss of GID function.

## Materials and methods

### Mammalian cell culture

HeLa, HEK-293, RPE and BJ human fibroblast cells were grown in NUNC cell culture dishes in Dulbecco's modified medium (DMEM) from Invitrogen supplemented with 10% FBS and 1% Penicillin-Streptomycin-Glutamine 100x (PSG, Life Technologies). HeLa Kyoto were kindly provided by Daniel Gerlich, RPE cells by Claus Azzalin, BJ cells were obtained from the American Type Culture Collection (ATCC), HeLa FRT cells were a kind gift of Stephen Taylor and HEK-293 FRT were bought from Life Technologies. All cell lines were passaged over a maximum period of 3 months and routinely tested negative for mycoplasma contamination. RNAi experiments were performed with Lipofectamine RNAiMAX (Invitrogen) with typically 20 nM siRNA for 72 hr. Stable cell lines were generated via engineered FRT-TetR cells using either the Flp-In system (Invitrogen) or lentivirus-mediated transduction. Clones were selected in 200 µg/ml Hygromycin and expression induced by adding 1 µg/ml Doxycycline for 24 hr. Transient overexpression of genes cloned into the pcDNA5/FRT/TO (Invitrogen) was achieved by transfection with Lipofectamine 2000 or 3000 (Invitrogen) following standard manufacturer's protocol. Cells were harvested for analysis after 48 hr. For complementation assays, stable expression of siRNA-resistant constructs was induced by adding 1 µg/ml Doxycycline simultaneous to RNAi transfection. For clonogenic assays, RPE MAEA or WDR26-KO cells were seeded into six-well dishes ($1 \times 10^3$ cells/well) 3 days post transduction. After 10 days of incubation, DMEM was removed, the colonies washed three times with PBS, stained with 0.25% crystal violet diluted in 80% methanol and quantified using ImageJ software. For specific drug treatment, cells were exposed to the proteasome inhibitor MG132 (final concentration 10 µM) or Cycloheximide (50 µM) for the indicated period of time at 37°C.

### Plasmids, CRISPR constructs and lenti virus production

The plasmids pTRIPZ, pMD2.G and psPAX2 vectors for lentivirus production were kindly provided by the laboratory of Prof. Wilhelm Krek. All genes in this study except RanBP9 were cloned from human cDNA using standard PCR protocols. The N-terminus of RanBP9 (bps 1–409) was synthesized by GeneArt™ (ThermoFisher) after a codon optimization step. The RNAi resistant constructs for WDR26 and the point mutants of the truncated Hbp1 (Hbp1$^{\Delta 1\text{-}10}$) were obtained by standard PCR-based mutagenesis. WDR26 and MAEA-specific guide RNAs were designed using the following online tools: crispr.mit.edu and crispor.tefor.net, respectively. Paired gRNAs were selected to improve the efficiency (*Wettstein et al., 2016*). The gRNAs were cloned into lentiCRISPRv2 vector

(Addgene), lentivirus particles were produced as described below and administered to RPE, HEK-293 and HeLa cells in pairs to transduce cells and generate stable KOs. For lentivirus production, 10 µg of various pTRIPZ or lentiCRISPRv2 constructs were co-transfected with 7 µg of psPAX2 and 3 µg of pMD2.G into HEK-293T cells (10 cm dish, 11 ml final volume) using Lipofectamine 2000 according to manufacturer's protocol. After 48 hr, 10 ml of the supernatant containing the viral particles was collected and sterile filtered (0.45 µm pores). Polybrene (Sigma) was added to the harvested virus at a final concentration of 8 µg/ml. Transduction of the virus into the respective cell line was performed by adding 500 µl of the virus solution to $2 \times 10^5$ cells in 2 ml medium (6-well plate), in the presence of polybrene (final conc. 8 µg/ml). After 2 days, cells were selected with Puromycin (final conc. 2 µg/ml). Expression from the pTRIPZ promoter was induced by addition of 1 µg/ml Doxycycline (Sigma).

## MTT assay

Proliferation rates and cytotoxicity of cells deleted for individual GID subunits was measured using the CellTiter 96 Non-Radioactive Cell Proliferation Assay (Promega). Briefly, $2.5 \times 10^3$ of the respective cells were seeded into 96-well plates and the MTT assay was performed after 48 hr according to manufacturer's instructions using a VersaMax microplate reader at 570 nm.

## RNA isolation and qRT-PCR

Total RNA was isolated using the RNeasy Mini Kit (QIAGEN). The reverse transcription was performed using random primers (Microsynth) and Superscript II Reverse Transcriptase (Invitrogen). For the qRT-PCR, cDNA was mixed with SYBR Green I Master (Roche) and the respective primer set before analysis with the LightCycler 480 (Roche Life Science).

## Immunoprecipitation, Western Blot and antibodies

For harvesting, cells were washed twice in PBS, scraped and pelleted at 1200 rpm for 1 min. Cell pellets for total protein analysis by Western Blotting and immunoprecipitation experiments were lysed in a buffer containing 20 mM Tris pH 8.0, 150 mM NaCl, 2 mM $MgCl_2$, 1 mM $CaCl_2$, 0.1% NP-40, 10% Sucrose, 1 mM DTT, 1 mM NaF, 1 mM PMSF, 0.5 µl/ml Pierce Universal Nuclease and Roche protease inhibitor cocktail). For analysis of total protein levels, cells were incubated in lysis buffer for 15 min before measuring the protein concentration. Subsequently, samples were boiled for 5 min at 95°C in 1x LDS (Novagen) and 10 mM DTT. For immunoprecipitation of SSH-tagged proteins, protein bait expression was induced by addition of 1 µg/ml of Doxycycline for 12 hr. Cell pellets of two confluent 15 cm dishes were resuspended in lysis buffer as described above and the chromatin sheared by passing the extract multiple times through a 26G needle. The extract was then cleared by centrifugation at 14 500 rpm at 4°C for 25 min, the protein concentration determined and adjusted. Subsequently, the lysate was incubated for 1.5–2 hr with 20 µl of pre-equilibrated Anti-HA antibody-coupled beads (Sigma). After binding, the beads were washed 3x with IP buffer, 3x with Wash buffer (50 mM Tris pH 8.0, 150 mM NaCl, 10% glycerol) and eluted from the affinity gel by addition of HA-peptide at a concentration of 1 mg/ml (ApexBio). Native GID particles were then boiled for 5 min at 95°C in 1x LDS (Novagen) and 10 mM DTT and proteins resolved by standard SDS-PAGE or NuPAGE 4–12% Bis-Tris Protein Gels (Invitrogen) before transfer onto Immobilon-P or Nitrocellulose transfer membranes (Millipore). Before incubation with the respective primary antibodies, membranes were blocked in 5% milk-PBST (MIGROS) for 1 hr. For protein detection primary antibodies detecting WDR26 (A302-244A, Bethyl Laboratories), Hbp1 (A5, sc-376831, Santa Cruz), Twa1 (5305, Prosci-Inc), MAEA (AF7288-SP, R and D Systems Europe Ltd), RanBP9 (A304-779A, Bethyl Laboratories), YPEL5 (ab103831, Abcam), HTRA2 (AF1458-SP, Novus Biologicals), LONP1 (A304-800A, Bethyl Laboratories), Cyclin D1 (ab134175, Abcam), Fbp1 (HPA005857, Sigma-Aldrich), Sin3a (A300-724A, Bethyl Laboratories), Cyclin A (H-432) (sc-751, Santa Cruz), Rb (554136, BD Pharmingen), Phospho-Rb (Ser780) (D59B7) (8180, Cell Signaling), Phospho-Rb (Ser807/811) (D20B12) XP (#8516, Cell Signaling), phospho-H3 (06–570, Upstate), HA (HA.11 antibody, MMS-101P, Covance), FLAG (M2, F3165, Sigma-Aldrich or F7425, Sigma-Aldrich), ubiquitin conjugates (P4D1, sc-8017, Santa Cruz), Tubulin (A-11029, Sigma-Aldrich), GADPH (G-8795, Sigma-Aldrich) and Vinculin (V9131, Sigma-Aldrich) were used. Secondary antibodies were goat anti-rabbit IgG HRP (170–6515, Bio-Rad), goat anti-mouse IgG HRP (170–6516, Bio-Rad), anti-mouse HRP TrueBlot (18-8817-33,

Bioconcept) and donkey anti-sheep HRP (A3415, Sigma-Aldrich). For re-probing, blots were stripped in a stripping buffer (10 mM Glycin pH 2.0, 1% SDS), washed several times in PBST and re-blocked.

## Protein expression and purification

The GID complex (FLAG-MAEA) and Hbp1-FLAG were reconstituted from Sf9 cells (Invitrogen) using the MultiBac expression system (Geneva Biotech). Expression cultures were seeded at a density of 2 × 10$^6$ cells/ml and infected with baculovirus stocks to obtain the full-length human GID complex or individual subcomplexes. Cells were harvested by centrifugation at 1200 rpm for 3 min and snap-frozen. For protein purification, insect cell pellets were resuspended in lysis buffer (50 mM Tris-HCl pH 8.0, 300 mM NaCl, 1 mM EDTA, 5 mM DTT, and Roche Complete Protease Inhibitor Cocktail tablet) and homogenized using a 15 ml Dounce homogeniser with 15 strokes. The lysate was cleared by centrifugation in a SS34 rotor at 17,000 rpm, at 4°C for 30 min. Protein extracts were then subjected to a single step FLAG purification. In brief, anti-FLAG-M2 affinity agarose beads (Sigma-Aldrich) were pre-equilibrated in lysis buffer and incubated with the extract for 1 hr, rotating at 4°C. Subsequently, the beads were washed 3x with each lysis- and wash buffer (50 mM Tris-HCl pH 8.0, 150 mM NaCl, 10% glycerol), and the proteins eluted with 1 mg/ml 3xFLAG peptide (APEXbio) in wash buffer, agitating for 1 hr at 4°C. E2 enzymes were expressed in bacteria as N-terminal GST-TEV fusion proteins. Expression was induced o/n with 0.1 mM IPTG at 18°C. Cells were harvested by centrifugation, resuspended in lysis buffer (50 mM Tris pH 8.0, 1% Triton-X, 350 mM NaCl, 1 mM EDTA, 5 mM DTT, Roche Complete Protease Inhibitor Cocktail tablet, Universal Nuclease (Pierce) and Lysozyme), sonicated and cleared from cell debris by centrifugation at 17,000 rpm for 30 min at 4°C. Subsequently, the protein extract was incubated with Glutathione Sepharose 4B (GE Healthcare) for 2 hr in the cold. The beads were then collected by centrifugation, washed 6x in lysis buffer, equilibrated with TEV cleavage buffer (50 mM Tris-HCl pH 8.0, 150 mM NaCl, 0.5 mM EDTA and 1 mM DTT) and incubated o/n with 6xHIS-TEV (homemade) to cleave off the GST-tag. The next day the eluate was incubated with Ni-NTA Agarose (QIAGEN) to remove the TEV protease from the purified E2 enzyme.

## Size exclusion chromatography and SEC-MALS

Purified full-length GID complexes or individual subcomplexes were subjected to size exclusion chromatography using Superose 6 10/300 or Superose 6 Increase 3.2/300 columns (GE Healthcare) in a buffer containing 50 mM Tris pH 8.0, 150 mM NaCl, 5% glycerol. Relative changes in the elution profile of the complexes were compared to protein standards and analyzed by SDS-PAGE followed by standard Coomassie- or Sypro-Ruby (Sigma-Aldrich) staining. Size exclusion chromatography followed by multi-angle light scattering (SEC-MALS) were performed in a buffer composed of 50 mM Tris-HCl, pH 8.0, 150 mM NaCl, 2 mM DTT using a Superose 6 10/300 analytical size exclusion chromatography column connected in-line to a miniDAWN TREOS light scattering and Optilab T-rEX refractive index detector (Wyatt Technology). Measurements were carried out at 20°C and the pentameric GID complex was used at 6 mg/ml (injected volume: 30 µl). Data analysis was performed using the software package provided by the manufacturer.

## In vitro ubiquitination

500 nM Hbp1-FLAG, 100 nM UbE1 (Boston Biochem), 1 µM E2 enzyme (Cdc34b, UbeH2, UbcH5c (Boston Biochem), UbcH5b, Ube2G2, Ube2E2), 50 µM Ubiquitin and 500 nM FLAG-GID complex were incubated at 37°C in a total volume of 50 µl reaction buffer (50 mM Tris pH 7.6, 3 mM ATP, 0.5 mM DTT, 10 mM MgCl$_2$). At the indicated time points, 10 µl of the reaction mixture was analyzed by SDS-PAGE and immunoblotting. For native HSS-GID particles we performed immunoprecipitation experiments and ubiquitination assays as described above with the following modifications; 25 µl of HSS-GID eluate (GID2 or GID5) was incubated with 100 nM UbE1, 1 µM E2 enzyme, 15 µM Ubiquitin and in a total of 50 µl E3 reaction Buffer. 10 µl samples were taken at variable time points and autocatalytic activity of the GID complex analyzed by Western Blotting.

## Analysis of murine primary hepatocytes and HEK-293 cells

Mice used for this study were on a C57Bl/6N background. Primary hepatocytes were isolated as previously described (*Mobin et al., 2016*). In short, livers were perfused with LiberaseTM (Roche) and

isolated hepatocytes were seeded at a density of $3 \times 10^5$ cells per well in BD Corning Primaria 6-well plates in DMEM low glucose supplemented with 10% FBS. After attachment, the hepatocytes were transfected with siRNAs using iMax reagent (Invitrogen) in Hepatozyme serum-free medium. HEK-293 cells were seeded at a confluency of 300.000 cells per well in six well plates and reverse transfected using iMax reagent in Optimem Medium. Cells were washed 3x with PBS 68–72 hr after transfection and glucose-free medium (DMEM A14430-01), supplemented with 20 μM forskolin, 0.22 g/l pyruvate and 2.24 g/l lactate, was added and cells were incubated for 5 hr. For the switch from gluconeogenic to glycolytic conditions the medium was changed to Hepatozyme (containing 25 mmol/l glucose) supplemented with 10 μg/ml insulin and incubated for 2, 3, 4, and 5 hr. For HEK-293, DMEM high glucose (25 mmol/l glucose) was used instead of Hepatozyme medium. Total RNA was isolated using Trizol reagent (Life Technologies) according to the manufacturer's protocol. DNAse treatment was performed using DNA free kit (Ambion) followed by reverse transcription using High Capacity cDNA Reverse Transcription Kit (Applied Biosystems). RT-PCR was carried out in an LC480 II Lightcycler (Roche) using gene-specific primers and Sybr Fast 2x Universal Master mix (Kapa). Results were normalized to 36b4 transcripts. Whole cell lysates for Western blotting were prepared in RIPA buffer (50 mM Tris-HCl pH 8.0, 150 mM NaCl, 1% NP-40, 0.5% sodium deoxycholate, 0.1% SDS and Complete Protease Inhibitor Cocktail (Roche)), incubated for 30 min on ice and centrifuged for 15 min at 12,000 rpm at 4°C. For protein detection blots were incubated overnight with primary antibodies detecting WDR26 (A302-244A-T), γ-Tubulin (T6557-2ML), FBP1 (ab109732), pAkt (S473) (Cell Signaling 4060), PCK1 (ab70358), and RanBP9 (ab205954). Blots were developed with a Fujifilm analyzer (LAS-4000) and signal intensities quantified using ImageJ.

## Mass spectrometry

For the identification of GID complex components and associated proteins, we performed AP-MS experiments starting with immunoprecipitation experiments as described above with the following modifications; after washing, the HA-beads were boiled for 5 min at 95°C in 1x LDS (Novagen) and loaded onto a NuPAGE 12% Bis-Tris Protein Gel (Invitrogen). Subsequently, the gel was stained with InstantBlue (Invitrogen), the protein bands cut into small pieces, washed 3x for 20 min with 200 μl 5 mM ABC/50% ACN and dehydrated for 10 min with 200 μl 100% ACN. For cysteine alkylation, the gel pieces were first reduced for 45 min at 56°C with 200 μl 10 mM DTT in 20 mM ABC, alkylated for 30 min with 200 μl 55 mM IAA in 20 mM ABC in the dark, followed by a 2x wash for 20 min with 200 μl 5 mM ABC/50% ACN. After that, the samples were dehydrated for 15 min with 200 μl 100% ACN and placed in the SpeedVac to remove any residual ACN. The dried gel pieces were rehydrated in 40 μl 12.5 ng/μl Trypsin in 20 mM ABC and incubated for 10 min at 22°C before adding 80 μl 20 mM ABC and incubating overnight at 37°C shaking at 750 rpm. The next day, the supernatant was collected into a fresh tube and the peptides eluted from the gel pieces with 100 μl 30% ACN/3% TFA for 30 min, 100 μl 80% ACN/0.5% AA ('Solvent B') for 30 min, 100 μl 100% ACN for 30 min. All the supernatants were pooled and the liquid reduced to 40 μl in the SpeedVac. For the peptide clean-up, $C_{18}$ Ziptips (Millipore) were first washed 10x in 100% ACN, followed by additional 10 washes with 3% ACN/0.2% FA before loading the peptides. The tips were then washed 10x with fresh 3% ACN/0.2% FA, eluted 4x with 10 μl 80% ACN, dried in the SpeedVac and resuspended in 4 μl 0.1% FA.

Peptides were analyzed by LC-MS/MS on a Q-Exactive plus (QE+) mass spectrometer (Thermo Scientific) equipped with a nano-electrospray ion source. Chromatographic separation of peptides was done with an EASY nano-liquid chromatography system (Proxeon) equipped with a 40 cm fused silica column with 75 μm inner diameter (New Objective), packed with Reprosil Pur C18 Aq 1.9 μm beads (Dr. Maisch) and heated to 50°C. 1 ul of the peptide mixtures was separated with a linear gradient from 5% to 35% acetonitrile in 90 min at 0.3 μl/min. Precursor scans were performed at a resolution of 70,000 at 200 $m/z$, and followed by acquiring twenty MS/MS spectra after high-energy collisional dissociation in the Orbitrap at a resolution of 17,500 at 200 $m/z$. An intensity threshold for fragmentation of 3.6e4 and a dynamic exclusion window of 30 s were used. The collected spectra were searched using Sequest HT within Proteome Discoverer (Thermo Scientific, version 1.4) against the Uniprot human proteome from November 2015. Trypsin was set as the digesting protease with the tolerance of 2 missed cleavages and fully-tryptic termini. The monoisotopic peptide and fragment mass tolerances were set to 10 ppm and 0.02 Da, respectively. Carbamidomethylation of cysteines (+57.0214 Da) was set as a fixed modification and oxidation of methionines (+15.99492 Da)

was set as a variable modification. Peptide spectrum matches were filtered using Percolator at a false discovery rate of 1%, determined using a reverse-sequence decoy database.

### Data analysis

Semi-quantitative densitometric analysis of immunoblots (acquired with a Fusion FX) and Sypro Ruby-stained gels (acquired with a Typhoon FLA9000) was performed using ImageJ software. For the AP-MS experiments, the raw data (spectral counts) was analyzed using the SAINT (*Choi et al., 2011*) scoring tool implemented within the Contaminant Repository for Affinity Purification (CRAPome) platform (*Mellacheruvu et al., 2013*). For calculation of SAINT scores, we followed workflow three using the following parameters: Fold change (FC-B); all controls, stringent. For enrichment analysis, the peptide spectrum matches for each protein were ranked by fold change versus the respective control immunoprecipitation.

### Immunofluorescence

For immunofluorescence, cells were grown on 12 mm coverslips, washed with PBS and then fixed in 4% PFA for 12 min at ambient temperature. After washing 3x with 0.01% Triton-X100 in PBS (PBS-TX100), cells were permeabilized in 0.5% NP40 in PBS for 5 min. The permeabilized cells were washed for an additional 3x in PBS-Tx100, incubated for 1 hr in the blocking buffer (5% FBS in PBS-TX100) and then for 1 hr in the primary antibody diluted in blocking buffer. Subsequently, cells were treated with the secondary antibody solution for 45 min (Alexa Fluor-conjugated anti-rabbit/or anti-mouse IgG (Life Technologies) and 0.2 µg/ml DAPI in PBS-TX100) prior to mounting using Immu-Mount (Thermo). The images were acquired on a fully automated inverted epifluorescence microscope (Ti-Eclipse, Nikon) using 60x oil objectives.

### Glucose production in primary hepatocytes

After transfection with GID- and control siRNA, glucose production in the media was measured by incubating primary hepatocytes for 6 hr in glucose-free media supplemented with forskolin (20 µM), pyruvate (0.22 g/L) and lactate (2.24 g/L). Cells were then cultured in Hepatozyme medium and incubated for 4 hr. Glucose content in the media was measured using a colorimetric assay (Chrystachem).

## Acknowledgements

We thank P Kimmig, P Wild, P Boersema and R Dechant for scientific input and critical reading of the manuscript, E Birkeland for quantification of clonogenic assays, C Wilson-Zbinden and M Gazor-pak for scientific support, the Krek lab at ETHZ for generously providing plasmids and cell lines, A Smith for editing the manuscript, and members of the Peter laboratory for critical discussions. Work was funded by ETHZ (MP, MS, PP), an ERC senior award (MP) and SNF research grants (MP, MS). FL is supported by the Human Frontier Science Program (HFSP; LT000376/2014).

## Additional information

### Funding

| Funder | Grant reference number | Author |
| --- | --- | --- |
| Human Frontier Science Program | LT- 000376/2014-L | Fabienne Lampert |
| Eidgenössische Technische Hochschule Zürich | | Paola Picotti<br>Markus Stoffel<br>Matthias Peter |
| Schweizerischer Nationalfonds zur Förderung der Wissenschaftlichen Forschung | | Matthias Peter<br>Markus Stoffel |
| European Research Council | | Matthias Peter |

The funders had no role in study design, data collection and interpretation, or the decision to submit the work for publication.

## Author contributions
Fabienne Lampert, Conceptualization, Formal analysis, Funding acquisition, Investigation, Methodology, Writing—original draft, Project administration, Writing—review and editing; Diana Stafa, Formal analysis, Investigation, Methodology, Writing—review and editing; Algera Goga, Formal analysis, Investigation, Writing—review and editing; Martin Varis Soste, Formal analysis, Investigation, Methodology; Samuel Gilberto, Investigation; Natacha Olieric, Methodology; Paola Picotti, Supervision; Markus Stoffel, Supervision, Funding acquisition, Writing—review and editing; Matthias Peter, Conceptualization, Supervision, Funding acquisition, Writing—original draft, Project administration, Writing—review and editing

## Author ORCIDs
Fabienne Lampert [iD] http://orcid.org/0000-0002-7560-4955
Diana Stafa [iD] http://orcid.org/0000-0002-4177-0197
Markus Stoffel [iD] http://orcid.org/0000-0003-1304-5817
Matthias Peter [iD] http://orcid.org/0000-0002-2160-6824

## Ethics
Animal experimentation: All animal experiments were approved by the Kantonale Veterinäramt Zürich.

## Decision letter and Author response
Decision letter https://doi.org/10.7554/eLife.35528.023
Author response https://doi.org/10.7554/eLife.35528.024

# Additional files
## Supplementary files
• Supplementary file 1 is related to the Materials and methods section and contains the lists of plasmids, siRNAs, qRT-PCR primers, guide RNAs, yeast strains and cell lines used in this study.
DOI: https://doi.org/10.7554/eLife.35528.018
• Transparent reporting form
DOI: https://doi.org/10.7554/eLife.35528.019

## Data availability
All data generated or analysed during this study are included in the manuscript and supporting files. Source data files contain the complete lists of mass spectrometry results and SAINT scores.

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
