## [Decision Letter]

Thank you for submitting your article "The multi-subunit GID/CTLH E3 ligase promotes proliferation and targets the transcription factor Hbp1 for degradation" for consideration by *eLife*. Your article has been reviewed by two peer reviewers, and the evaluation has been overseen by Ivan Dikic as the Reviewing Editor and Senior Editor. The following individual involved in review of your submission has agreed to reveal his identity: Jakob Nilsson (Reviewer #2).

The reviewers have discussed the reviews with one another and the Reviewing Editor has drafted this decision to help you prepare a revised submission.

General review:

Following their interest in the regulation and function of Cul4-based CRLs, the authors question here the role of WDR26 as a potential substrate adaptor. Their interaction proteomics results suggest that instead, WDR26 co-purifies with human orthologues of the yeast GID complex, in accord with the fact that WDR26 displays some homology to yeast Gid7. The GID complex is a yeast multisubunit RING-based E3 involved in the degradation of gluconeogenic enzymes. The structural conservation of its components in human (aka CTHL complex) led to suggestions that it may also be an E3 but this had not been experimentally demonstrated. It is really exciting to finally have some functional data on this complex in human cells.

The authors convincingly show that this complex possesses an intrinsic E3 activity. The recombinant expression and purification of this multisubunit complex is a tour de force allowing them to begin to address the composition/stoichiometry/interactions within the complex, its E3 ligase activity and its mode of substrate recognition.

The authors claim that murine GID does not target the gluconeogenic enzyme FBPase, but I find it hard to conclude this with the existing data. If this were true, this is not so surprising anyway, given the differences in cellular physiology that the authors highlight very well in the discussion. However, they obtained convincing evidence using an unbiased method (IP-MS) and further confirmation that the transcription factor and tumour suppressor, Hbp1, is a GID interactant, and a substrate. The latter part is particularly well performed and solid, with siRNA or CRISPR of multiple subunits, on endogenous Hbp1 protein, measurement of protein levels but also stability, with complementation with siRNA-resistant constructs, etc. in vitro ubiquitylation of Hbp1 by hGID is also provided. Recently published data showed that the yGID is a Pro-targeted N-end rule E3 (Chen et al., 2017, cited). Interestingly, Hbp1 does not seem to follow the Pro-N-end rule, and accordingly, yFBP1 is not an in vitro substrate of hGID. In order to further improve their conclusions the authors are invited to address the following:

Major points:

1) CRISPR-based KO of two subunits, MAEA and WDR26, leads to defects in proliferation reminiscent of previous work on other GID subunits (cited in the manuscript).The only thing missing is the final proof that cell proliferation defects caused by GID knock-down/out are caused by the observed increase in Hbp1 expression. The authors should try to address this issue, even though this may be difficult since the proliferation defect may also be multifactorial with additional, unknown substrates beyond Hbp1 as noted in the Discussion.

2) It does not appear clearly which subunit was silenced in Figure 3F and in Figure 3—figure supplement 1E ("GID siRNA": which GID?). Anyway, this experimental setup cannot allow to conclude that mFBP1 is not a substrate of murine GID since mFBP1 appears not to be degraded upon switching to a glycolytic mode even in WT hepatocytes (despite the drop in transcript level). To claim this, the authors should use a different setup (different cell type, different way to operate a metabolic switch), or a different readout (what about PEPCK, MDH etc.). Otherwise, they should tone down and claim that they have no evidence that the GID is involved in the degradation of gluconeogenic enzymes, and not be so decisive about the lack of involvement in this process. The model and its legend are far-reaching, there is no definitive demonstration that gluconeogenesis is not affected, nor of which subunit interacts with Hbp1, nor that some subunits may target the GID at multiple places in the cell etc.

3) Figure 2B suggests that the authors are looking at a mix of complexes, it is not convincing that this is a stable complex looking at the elution profiles of the individual subunits. The authors should run SEC-MALS to address this issue. Also looking at staining of Figure 2A this could also indicate differences in stoichiometry.

---

## [Author Response]

Major points:1) CRISPR-based KO of two subunits, MAEA and WDR26, leads to defects in proliferation reminiscent of previous work on other GID subunits (cited in the manuscript).The only thing missing is the final proof that cell proliferation defects caused by GID knock-down/out are caused by the observed increase in Hbp1 expression. The authors should try to address this issue, even though this may be difficult since the proliferation defect may also be multifactorial with additional, unknown substrates beyond Hbp1 as noted in the Discussion.

We have initiated multiple attempts to assess the relative contribution of Hbp1 to the proliferation phenotype of GID-KO RPE cells already before we submitted this manuscript. First, we performed simultaneous depletion of MAEA and Hbp1 by siRNA in RPE cells. The consecutive phenotypical analysis however was hampered by inefficient downregulation of both Hbp1 and GID subunits. Second, we aimed to downregulate Hbp1 by siRNA in MAEA- or WDR26-KO RPE cells. However, these early GID-KO cells are severely compromised and did not tolerate additional transfections. Third, we tried to delete Hbp1 by CRISPR in an already established MAEA-KO RPE cell line but were unable recover a viable clone with the desired genotype. Finally, we created Hbp1-KO RPE cells but failed to isolate a viable clone where Hbp1 is fully deleted. Nevertheless, we processed partial Hbp1-KO cells and additionally deleted WDR26 and MAEA. Unfortunately, we were not able to obtain the required cells due to deleterious cytotoxic effects. Thus, we are currently not able to answer this pressing question. We are now constructing auxin-degradable knock-in RPE cell lines to acutely remove Hbp1 and GID-subunits. In parallel, we started to screen for chemical compounds that specifically inhibit the enzymatic activity of the hGID ligase. We are confident that these efforts will be of great benefit to learn more about Hbp1 and other GID substrates, but find these experiments clearly beyond the scope of the current manuscript.

2) It does not appear clearly which subunit was silenced in Figure 3F and in Figure 3—figure supplement 1E ("GID siRNA": which GID?).

A pool of siRNAs that target the subunits MAEA, Rmnd5a, RanBP9, Armc8, GID4, WDR26 and Twa1 has been used for these experiments (collectively termed “GID siRNA). We have added this information to the text. The specific oligos are listed in the supplementary information.

Anyway, this experimental setup cannot allow to conclude that mFBP1 is not a substrate of murine GID since mFBP1 appears not to be degraded upon switching to a glycolytic mode even in WT hepatocytes (despite the drop in transcript level). To claim this, the authors should use a different setup (different cell type, different way to operate a metabolic switch), or a different readout (what about PEPCK, MDH etc.). Otherwise, they should tone down and claim that they have no evidence that the GID is involved in the degradation of gluconeogenic enzymes, and not be so decisive about the lack of involvement in this process. The model and its legend are far-reaching, there is no definitive demonstration that gluconeogenesis is not affected, nor of which subunit interacts with Hbp1, nor that some subunits may target the GID at multiple places in the cell etc.

We repeated the experiment pointed out by the reviewer in Hepa1-6 (cell line with very low FBP1 protein levels) and in HEK-293 cells. We also immunoblotted for additional proteins regulating the gluconeogenic pathway, including PCK1 as suggested. Consistent with our results in primary hepatocytes, we were not able to detect a substantial change in both FBP1 and PCK1 protein levels when shifting cells from gluconeogenesis to glycolysis in GID-depleted cells (revised Figure 3, new panels F and G). Likewise, increasing the incubation time when shifting cells to glycolysis did not change PCK1 and FBP1 levels. To confirm that the cells indeed switch from gluconeogenesis to glycolysis we immunoblotted for phosphorylated Akt(Ser473)^1^. Moreover, RT-qPCR data from control and GID-depleted primary hepatocytes and HEK-293 cells showed that FBP1 and PCK1 are as expected transcriptionally regulated, but this regulation is not dependent on the murine GID E3 ligase^2,3^.

**Author response image 1. respfig1:** Murine GID is dispensable for the regulation of gluconeogenic enzymes FBP1 and PCK1. A, B. Western blot analysis of control- and GID siRNA-treated primary mouse hepatocytes and HEK-293 cells shifting from gluconeogenesis (GNG) to glycolysis (GLY) over the indicated period of time. C, D. qRT-PCR of PCK1 and FBP1 transcripts in HEK-293 and primary hepatocytes in control and GID-depleted cells during a metabolic shift from GNG to GLY. Data is shown as mean fold change relative to control ± SD, n = 3.

Finally, we have performed additional experiments showing that the murine GID E3 ligase complex is not playing a significant role in negatively regulating the levels of gluconeogenic enzymes. First, glucose production in control and GID-depleted cells is comparable (see Author response image 2). If the GID complex would negatively regulate gluconeogenic enzymes in mammalian cells, we expect that loss of GID activity would affect the ability of cells to produce glucose when cells are shifted from gluconeogenesis to glycolysis. Second, unlike *in S. cerevisiae,* knockdown of GID subunits does not alter glucose levels when cells are shifted from gluconeogenesis to glycolysis (see Author response image 2).

**Author response image 2. respfig2:** Murine GID does not affect glucose output in primary hepatocytes during a metabolic switch. A. Glucose production in primary hepatocytes following a 6 h incubation in glucose-producing medium. Data expressed as mean mg/dl of glucose per μg protein relative to control ± SD, n=6. B. Glucose levels in the medium of primary hepatocytes after a 4 h shift from gluconeogenesis to glycolysis. Data presented as mean fold change relative to control ± SD, n=3.

Taken together, these new experiments support our previous conclusion and collectively demonstrate that FBP1 and PCK1 levels are not regulated by the murine GID E3 ligase complex. We have thus revised Figure 3 and Figure 3—figure supplement 1 accordingly (new panels F and G, respectively).

3) Figure 2B suggests that the authors are looking at a mix of complexes, it is not convincing that this is a stable complex looking at the elution profiles of the individual subunits. The authors should run SEC-MALS to address this issue. Also looking at staining of Figure 2A this could also indicate differences in stoichiometry.

We agree that this aspect needs further attention and thus subjected the purified pentameric GID subcomplex composed of Twa1, MAEA, Rmnd5a, WDR26 and RanBP9 as suggested to SEC-MALS analysis to determine its molecular mass in solution. SEC-MALS yielded two major peaks; the first peak, with an average molecular mass of the 543 kDa, is consistent with the formation of a GID dimer (calculated molecular mass of the pentamer: ~ 285 kDa). As shown in the Sypro-stained SDS-PAGE in Figure 2B the complex separates on the Superose 6 column into a second stable trimeric subcomplex consisting of Twa1, MAEA and Rmnd5a and dimeric Twa1 which in SEC-MALS corresponds to a broad second peak ranging from ~ 155 kDa to 60 kDa in agreement with the predicted molecular weight of the respective proteins and constitutive dimerization of Twa1 (calculated molecular mass of the Twa1: ~ 27 kDa). These additional data further expands the biochemical GID analysis and are now included in the revised Figure 2 (new panel D).

References:

1) Puigserver, P. et al. Insulin-regulated hepatic gluconeogenesis through FOXO1-PGC-1alpha interaction. Nature 423, 550-555, doi:10.1038/nature01667 (2003).

2) Cassuto, H. et al. Involvement of HNF-1 in the regulation of phosphoenolpyruvate carboxykinase gene expression in the kidney. FEBS letters 412, 597-602 (1997).

3) Yang, J., Kalhan, S. C. and Hanson, R. W. What is the metabolic role of phosphoenolpyruvate carboxykinase? The Journal of biological chemistry 284, 27025-27029, doi:10.1074/jbc.R109.040543 (2009).